

# Inclusion of the ECMWF ecRad radiation scheme (v1.5.0) in the MAR model (v3.14), regional evaluation for Belgium and assessment of surface shortwave spectral fluxes at Uccle observatory

Jean-François Grailet[1], Robin J. Hogan[2], Nicolas Ghilain[1,3], Xavier Fettweis[1], and Marilaure Grégoire[1]

[1]University of Liège, Liège, BE
[2]European Centre for Medium-Range Weather Forecasts, Reading, UK
[3]Royal Meteorological Institute of Belgium, Uccle, BE

**Correspondence:** Jean-François Grailet (Jean-Francois.Grailet@uliege.be)

**Abstract.** The MAR model (*Modèle Atmosphérique Régional*) is a regional climate model used for weather forecasting and climate studies over several continents, including polar regions. To simulate how solar and Earth's infrared radiation propagate through the atmosphere and drive climate, MAR uses the Morcrette radiation scheme. Last updated in the 2000's, this scheme is no longer maintained and lacks of flexibility to add new capabilities, such as computing high resolution spectral fluxes.

This paper presents version 3.14 of MAR, an update that allows MAR to run with ecRad, the latest radiation scheme provided by the European Centre for Medium-range Weather Forecast (ECMWF). Operational in the ECMWF's Integrated Forecasting System (IFS) since 2017, ecRad was designed with modularity in mind and is still in active development.

We evaluate the updated MAR by comparing its outputs over 2011–2020 for Belgium to gridded data provided by the Royal Meteorological Institute of Belgium (RMIB) and by the EUMETSAT Satellite Application Facility on Land Surface Analysis.
Several sensitivity experiments have been carried out to find the configuration achieving the most balanced radiative budget as well as to demonstrate the updated MAR outperforms its former configuration. Moreover, a MAR simulation running ecRad with high resolution ecCKD gas-optics models has been conducted to produce spectral shortwave fluxes, which are compared to ground-based spectral measurements captured by the Royal Belgian Institute for Space Aeronomy (BISA) at Uccle (Belgium; 50.797 °N, 4.357 °E) in the 280–500 nm range from 2017 to 2020. Finally, as a first application of spectral shortwave fluxes
computed by MAR running with ecRad, a method for predicting UV indices is described and evaluated.

## 1 Introduction

The MAR model is a regional atmospheric model that was initially designed by Gallée and Schayes (1994) to study the climate of various regions at high resolution (typically ranging from 5 to 25 km) over periods of time covering a few years up to several decades (Fettweis et al., 2013b). Throughout its decades of development, MAR has been tuned for polar areas in particular,
notably by coupling it with a snow model (De Ridder and Gallée, 1998; Gallée et al., 2001). This development allowed MAR users to study the surface mass balance of the Greenland ice sheet starting from the early 2000's (Gallée and Duynkerke, 1997; Lefebre et al., 2003, 2005). Since then, MAR has been run to estimate the future impact of the Greenland ice sheet on sea level



rise (Fettweis et al., 2013a) and to predict the evolution of its surface temperatures (Delhasse et al., 2020; Hanna et al., 2020). MAR is also used to study the Antarctic ice sheet (Amory et al., 2021; Kittel, 2021) as well as the evolution of the precipitation

regime of various regions, such as western and equatorial Africa (Gallée et al., 2004; Doutreloup, 2019) and central Europe (Wyard et al., 2020; Ménégoz et al., 2020).

A key component of the MAR model is its radiative transfer scheme, or radiation scheme. This module simulates how both shortwave (solar) and longwave (Earth's infrared) radiation propagate through the atmosphere and over the surface, with the latter resulting from the Earth's surface being heated by the former. To accurately simulate the transfer of radiative

energy, a radiation scheme must take into account all optically active components within the atmosphere that either reflect or absorb and scatter radiation (if not both), such as the Earth's surface (and its albedo), greenhouse gases, aerosols, and clouds. Having an accurate radiation scheme is crucial for a climate model, as how the radiative energy flows within the atmosphere and over the surface is what regulates the Earth's surface temperature. MAR up to version 3.13 uses the Morcrette (1991, 2002) radiation scheme, consisting of two separate schemes, respectively for shortwave (solar) radiation and longwave

(Earth's infrared) radiation. This scheme was notably used in the ERA-40 reanalysis (Uppala et al., 2005) and is still used in several regional climate models (Jacob et al., 2012; Hourdin et al., 2013).

Developed until the early 2000's, the Morcrette scheme is no longer maintained. Moreover, it has become difficult to update to improve or expand its capabilities due to the lack of modularity of its source code, partly because it pre-dates Fortran 2003, which improved derived types and introduced object-oriented programming in the language (Reid, 2007). The difficulty of

updating the Morcrette scheme hinders potential improvements for the MAR model, such as producing spectral shortwave fluxes to simulate snow albedo more accurately. Updating the radiation component of MAR also constitutes an opportunity to fix its known radiative biases: MAR running with Morcrette is known for overestimating downward shortwave radiation and underestimating downward longwave radiation, as evidenced by Fettweis et al. (2017), Delhasse et al. (2020), Wyard et al. (2018) and Kittel et al. (2022). Up to version 3.13 (Fettweis et al., 2023), MAR partially mitigates this issue by tuning the

outputs of the Morcrette scheme to slightly compensate known heat fluxes biases, depending on the simulated region, allowing MAR to simulate a correct near-surface temperature.

This paper discusses the inclusion in the MAR model of a new radiative transfer scheme: the ecRad radiation scheme (Hogan and Bozzo, 2018), the latest radiative transfer scheme developed by the ECMWF. Since 2017, ecRad is used as the radiative transfer component of the operational weather forecast model of the ECMWF, the IFS. The key feature of ecRad

is its modularity: its architecture allows users to change independently, among others, the description of optical properties with respect to clouds, greenhouse gases and aerosols, or the radiation solver (Hogan and Bozzo, 2018). Thanks to recent development, the ecRad scheme is now also compatible with high resolution gas-optics models built by the ecCKD tool of the ECMWF (Hogan and Matricardi, 2022) and even capable of outputting spectral shortwave radiative fluxes. Including the ecRad radiation scheme in the MAR model can therefore offer new research opportunities for the latter: for instance, MAR

may produce high resolution spectral fluxes in the photosynthetically active region of the solar spectrum to force biosphere and ocean biogeochemical models.



**Table 1.** Simplified timeline of the radiation schemes (and their main components) developed by the ECMWF and used in the IFS. A more complete timeline (with more components and code updates) covering the 2000–2017 period can be found in Hogan and Bozzo (2018). The bold elements correspond to the radiation scheme (and components) still used by MAR up to version 3.13.

|  | 2000 | 2002 | 2007 | 2017 | 2022 |
|---|---|---|---|---|---|
| Name | **Morcrette** |  | McRad | ecRad |  |
| Shortwave | Custom, 4 bands | **Custom, 6 bands** | RRTM-G (1997), 14 bands |  | RRTM-G (1997), 14 bands **or** ecCKD (2022), customizable |
| Longwave | **RRTM-G (1997), 16 bands** |  |  |  | RRTM-G (1997), 16 bands **or** ecCKD (2022), customizable |
| Solver | **Custom** |  | McICA (2003) | McICA (2003) **or** Tripleclouds (2008) **or** SPARTACUS (2016) |  |

The paper is organized as follows. Section 2 first presents the ecRad radiation scheme by briefly reviewing its development history and key features, including its recent updates with respects to spectral resolution, and by discussing how these features can benefit the MAR model. Section 3 subsequently details the changes and additions brought to MAR to interface it with

ecRad and take advantage of its improved representation of optically active components and improved spectral resolution. Section 4 then evaluates the updated MAR model (version 3.14) embedding ecRad for Belgium by simulating the 2011–2020 decade and comparing the output variables of MAR, and radiative fluxes in particular, to gridded reference data based on ground observations and/or satellite measurements, provided by the RMIB and by the EUMETSAT Satellite Application Facility on Land Surface Analysis. To highlight the benefits of including ecRad in MAR, Section 5 compares spectral shortwave fluxes

produced by MAR v3.14 to spectral observations recorded by the BISA at Uccle observatory (Belgium; 50.797 °N, 4.357 °E) from 2017 to 2020 and discusses a first application of such fluxes, consisting of predicting UV indices. Finally, Section 6 concludes this paper by summarizing its contributions, possible improvements and opportunities.

## 2 The ecRad radiation scheme

### 2.1 Development history

The ecRad radiation scheme is the latest milestone within several decades of development to improve the fidelity of the radiative transfer scheme used within ECMWF's operational weather prediction model, the IFS. Starting from the 1990's, the IFS used the radiation scheme of Morcrette (1991), which itself has gone through several updates up to the early 2000's to incorporate various advances in modeling. One major update of the Morcrette scheme was the inclusion, in 2000, of the Rapid Radiative Transfer Model for GCMs (RRTM-G) by Mlawer et al. (1997), a correlated-k model for gas absorption. At the time, RRTM-

G was used for longwave radiation only and significantly improved the estimation of surface downward longwave radiation compared to contemporary models (Morcrette, 2002). For shortwave radiation, the Morcrette scheme used a custom gas-optics



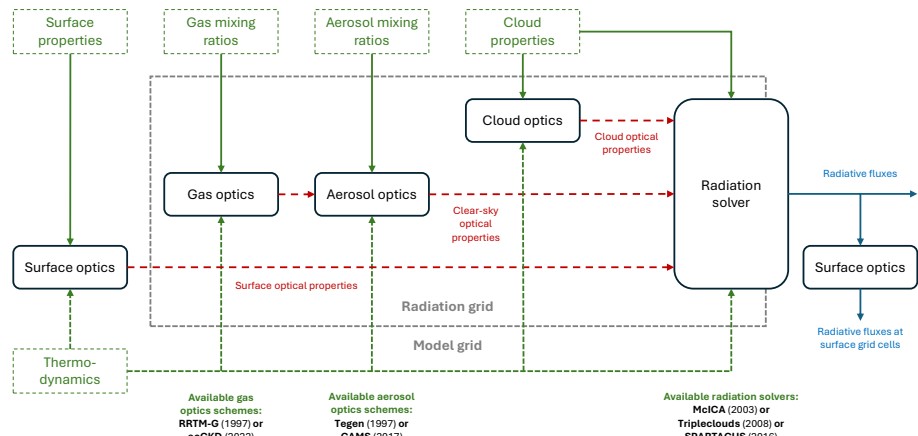

**Figure 1.** A high-level view of the architecture of the ecRad radiation scheme, inspired by Fig. 1 in Hogan and Bozzo (2018). The rounded and dashed square boxes correspond respectively to ecRad (interchangeable) components and to ecRad inputs. The thermodynamics box, an input of all ecRad components represented by short dashed arrows, corresponds to the description of temperature and pressure at each layer interface and within each layer. Long dashed arrows model the inner ecRad data structures. Finally, the background dashed box accounts for the interpolation steps used to go from the model to the radiation grid and vice versa.

scheme with 4 spectral bands in 2000, soon increased to 6 spectral bands in 2002 (Hogan and Bozzo, 2018). This version was notably used in the ERA-40 reanalysis (Morcrette, 2002; Uppala et al., 2005).

In 2007, the Morcrette scheme in the IFS was replaced by McRad (Morcrette et al., 2008), an improved radiative transfer
scheme. In addition to a better representation of surface albedo, this new scheme featured RRTM-G for both shortwave and longwave radiation (respectively with 14 and 16 spectral bands) and the Monte-Carlo Independent Column Approximation scheme (McICA) by Pincus et al. (2003), a radiation solver that simulates clouds with a stochastic generator. Though the McRad radiation scheme was a notable improvement over the Morcrette scheme, its source code was not designed for modularity, making it difficult to include new schemes that act either as alternatives or as improvements over those already provided.

The ecRad radiation scheme by Hogan and Bozzo (2018), which became operational in 2017, provides a more modern modular structure, facilitating inclusion of new advances in atmospheric modeling. Organized in several components through which data flow via dedicated structures, ecRad allows users to swap one scheme with another in each component without breaking the flow of the whole radiative transfer scheme, as illustrated by Fig. 1. Moreover, ecRad manages most of the data required by its inner components with NetCDF files, while the Morcrette scheme still used in MAR relies on hard-coded data,
notably for ozone mixing ratios.

Thanks to its flexibility, ecRad provides, among others, several interchangeable schemes to solve radiation equations. In addition to McICA (Pincus et al., 2003), already used by McRad, ecRad also provides the Triplecouds radiation solver by Shonk and Hogan (2008), which models horizontal cloud inhomogeneity by representing three regions within each grid cell.





The more recent SPARTACUS (SPeedy Algorithm for Radiative TrAnsfer through CloUd Sides) scheme is another proposed
alternative, capable of simulating 3-D cloud longwave radiative effects (Schäfer et al., 2016; Hogan et al., 2016).

The treatment of the optical properties of gases is another major sub-component of a radiation scheme that may be exper-
imented with. Both the IFS and the MAR model have used the RRTM-G gas-optics scheme for Earth's longwave radiation
since the 2000's, with the former also using it for shortwave radiation starting from 2007 with the replacement of the Morcrette
scheme by McRad. A promising alternative to this state-of-the-art solution is ecCKD (Hogan and Matricardi, 2022), an open
source tool from the ECMWF that builds fine-tuned gas-optics models via the correlated k-distribution method by Goody et al.
(1989), these models being managed with NetCDF files for convenience. Such a tool can be used to tailor high resolution
gas-optics models for specific applications while remaining computationally efficient.

## 2.2  Opportunities for the MAR model

The MAR model is a Regional Climate Model (RCM) consisting of an atmospheric module (Gallée and Schayes, 1994; Gallée,
1995) coupled with SISVAT (Soil–Ice–Snow–Vegetation–Atmosphere–Transfer), a one-dimensional surface transfer scheme
(De Ridder and Gallée, 1998; Gallée et al., 2001). This coupling makes MAR a suitable tool to study polar ice sheets, notably
in Greenland (Gallée and Duynkerke, 1997; Lefebre et al., 2003, 2005). To compute shortwave and longwave radiative fluxes
for its atmospheric module, MAR version 3.13 still uses a late build of the Morcrette scheme (Morcrette, 1991) from 2002 that
includes RRTM-G (Mlawer et al., 1997) for longwave radiation (cf. Table 1). An exhaustive description of the current MAR
model is provided by Gallée et al. (2013).

Used by the ECMWF from the early 1990's to the mid-2000's (Morcrette et al., 2008), the Morcrette scheme is arguably
too old to continue using it as the radiation scheme of MAR. First of all, it is no longer actively maintained and does not
benefit from Fortran language updates from the 2000's that would improve its modularity, such as improved derived types and
object-oriented programming mechanisms (Reid, 2007), therefore hindering the implementation of new or improved features.
Furthermore, some of its native characteristics are now outdated. For instance, the Morcrette scheme still uses the monthly cli-
matology by Tegen et al. (1997) for tropospheric aerosols, which has little reason to be preferred over more modern climatolo-
gies, and in particular those using the Copernicus Atmospheric Monitoring Service (CAMS) aerosol specification (Flemming
et al., 2017; Bozzo et al., 2017). Finally, another motivation for considering an update of the radiation component of MAR
is to eventually improve the radiative fluxes biases. Indeed, MAR with Morcrette is known to have an imbalance between
shortwave and longwave radiative fluxes, the former being typically overestimated while the latter is underestimated (Fettweis
et al., 2017; Delhasse et al., 2020; Wyard et al., 2018; Kittel et al., 2022).

Including the ecRad radiation scheme in the MAR model comes with multiple benefits. Replacing the aging Morcrette
scheme by ecRad in MAR is not only a way to modernize MAR itself and an opportunity to fix its known imbalance between
shortwave and longwave radiation, but also a way to expand MAR capabilities without requiring any major change in the MAR
source code. In particular, the latest version of ecRad (Hogan, 2024a) allows the computation of spectral shortwave radiative
fluxes into user-defined bands. Such a development offers new opportunities for the MAR model: for instance, the spectral





fluxes MAR could output may be used as forcings for other models or as a mean of coupling MAR with them. Models that could use such forcings include biogeochemical models designed to study marine ecosystems (Lazzari et al., 2021b).

Another recent key development of ecRad that further motivates this new kind of application of MAR is the ability of
ecRad to use high resolution gas-optics models built with the ecCKD tool (Hogan and Matricardi, 2022) within its gas optics component (cf. Fig. 1) in place of the classical RRTM-G scheme by Mlawer et al. (1997). The resulting increase in spectral resolution is especially important. To benefit from the spectral information provided with the new generation of satellites like Sentinel, modern ocean biogeochemical models rely on a refined radiative transfer module that manages spectral bands extending over a few dozens of nanometers, e.g. 25 nm (Lazzari et al., 2021b, a). This module needs to be forced at the air-sea
interface by spectral radiative fluxes, in particular in the Photosynthetically Active Range (PAR, 400–700 nm) but also in the UV and near-infrared ranges. Meanwhile, the (fixed) spectral bands of the RRTM-G gas-optics scheme in ecRad include one band from 442 to 625 nm, which amounts to almost two thirds of the PAR, making the RRTM-G gas-optics scheme a poor candidate to compute spectral radiative fluxes in such a range.

By swapping RRTM-G with high resolution ecCKD gas-optics models, ecRad can output equally high resolution spectral
shortwave fluxes, spanning over one or several dozens of nm, assuming the bands requested by the user are not finer in resolution (though this would only decrease the accuracy of the produced fluxes). Therefore, embedding ecRad within MAR and configuring it to run with ecCKD gas-optics models opens the door for new future applications of the MAR model as well as for improvements of MAR itself. For instance, high resolution spectral shortwave fluxes produced by ecRad/ecCKD may be used to implement a spectral snow albedo in MAR.

## 145  3  Inclusion of ecRad in MAR

The ecRad radiation scheme (version 1.5.0) is written in Fortran 2003 and consists of about 16,000 lines of code without counting the source code of the RRTM-G scheme (Hogan and Bozzo, 2018). It is also an open source software that comes with excerpts from the IFS source code to show how the IFS initializes then runs ecRad throughout a simulation. While ecRad itself did not require any modification to be embedded in MAR, the source code of MAR needed some adjustements to fully take
advantage of the improved representation of optically active components of ecRad as well as its improved spectral resolution.

### 3.1  Updated greenhouse gas and aerosol forcings

Due to the development history of the ecRad radiation scheme and the IFS (cf. Sect. 2.1), the example subroutines from the IFS (Hogan, 2024a) share many similarities in terms of interface with the subroutines calling the Morcrette scheme (Morcrette, 1991, 2002) in MAR. Among others, the input variables of the IFS subroutine that calls ecRad include the same description
of pressure and temperature profiles as for the Morcrette scheme, and this also holds true for several water species and surface variables (e.g., albedo, emissivity in the longwave or land/sea mask). However, ecRad natively offers more flexibility regarding greenhouse gases and aerosols. Indeed, for each greenhouse gas or aerosol, ecRad expects volume or mass mixing ratios for each pressure layer of each air column of the encompassing model grid. The number of aerosol species taken into account by



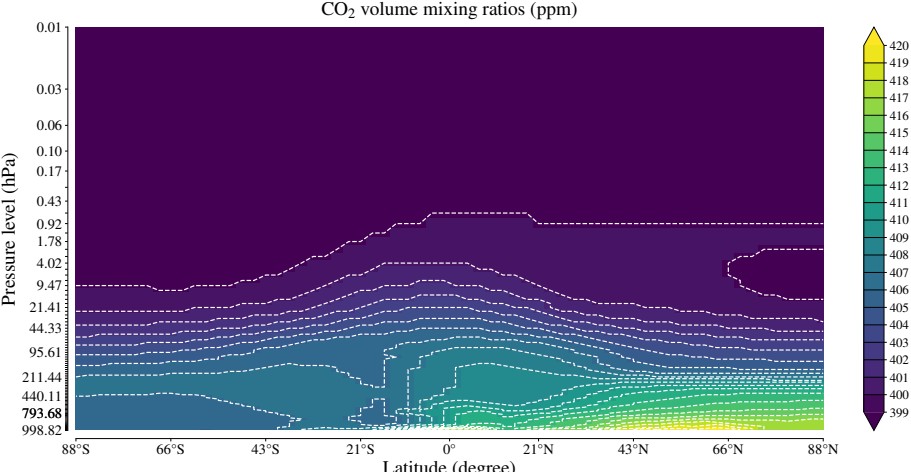

**Figure 2.** Longitude-averaged volume mixing ratios of $CO_2$ for a meridional transect of the Earth's atmosphere for January 2019 (SSP585).

ecRad can also be freely tuned. In comparison, the Morcrette scheme as used by MAR expects a single average mixing ratio
for most greenhouse gases and restricts aerosols to six species, as in the climatology of Tegen et al. (1997).

Therefore, to make the most of ecRad radiation scheme, the greenhouse gases and aerosols forcings of the MAR model were updated, re-using climatological data used in the IFS (cycle 46r1). The greenhouse gas forcings consist of twelve monthly 2-D grids providing longitude-averaged mixing ratios for a meridional transect of the Earth's atmosphere, based on reanalyses over the 2000's. For consistency's sake, the initial volume mixing ratios should be scaled to match the average surface mixing
ratios of the year simulated by MAR, these averages depending on either historical records or future climate scenarios. Such averages can be found in time series of the average surface volume mixing ratio of each greenhouse gas and for each year (from 0 to 2500) according to the Intergovernmental Panel on Climate Change's (IPCC) typical scenarios, the Representative Concentration Pathways (RCP) (Van Vuuren et al., 2011) and Shared Socioeconomic Pathways (SSP) (O'Neill et al., 2016). Given a climate scenario time series, the consecutive average surface mixing ratios corresponding to the period covered by
the initial forcings are used to compute a period average. Then, the average surface mixing ratio corresponding to the year MAR should simulate is divided by the period average to compute a scaling factor. This scaling factor is then applied on the forcings for the month MAR should simulate. Finally, the scaled forcings are interpolated in the MAR grid and smoothed by averaging with respects to the horizontal axis. This averaging mitigates the jumps in values that are due to the low resolution of the forcings with respects to latitude, as the aforementioned forcings define a total of 64 air columns between both poles in
the meridional transect, with a latitude degree difference between adjacent air columns ranging from 2.8 to 3.2°.

As ecRad is compatible with the CAMS aerosol specification, consisting of 11 hydrophilic or hydrophobic aerosol species (Flemming et al., 2017; Bozzo et al., 2017), the MAR model was also updated to provide such forcings to ecRad. Their preparation is essentially the same as for greenhouse gas forcings, the main difference being that CAMS data provided by the ECMWF (Hogan, 2024b) consists of monthly 3-D grids covering the global Earth's atmosphere. However, with 61 air columns





between poles in each of the 120 meridional transects w.r.t. longitude, only a few air columns from the initial forcings will be covering the high resolution MAR grid. Therefore, for simplicity's sake, the longitude coordinates of the MAR grid are averaged to select the meridional transect within the 3-D CAMS data that is the closest in longitude to the MAR grid. The subsequent steps to prepare the aerosol forcings are then identical to those used to prepare the greenhouse gases forcings.

### 3.2    Stratospheric layers on top of MAR grid

As the MAR vertical discretization is usually tuned to simulate the whole troposphere and the lower part of the stratosphere (i.e., slightly above the pressure value of 90 hPa), the impact of the stratosphere on radiative fluxes cannot be properly simulated with the MAR grid alone. This limitation is especially problematic for ozone, which is typically found in greater concentrations in the stratosphere, with then significant absorption of incoming solar radiation above 90 hPa, particularly in the ultraviolet range (Hogan and Matricardi, 2020). As one of the motivations for replacing Morcrette by ecRad in MAR is to have a better
and finer spectral representation of shortwave radiation (cf. Sect. 2.2), it is crucial to ensure the spectral effect of ozone is properly represented.

However, adding stratospheric layers to the MAR grid may be counterproductive given the extra computational cost brought by these additional layers. Indeed, as the stratosphere has little water vapor, stratospheric layers have little relevance for the physical processes typically studied with the MAR model, such as precipitation, (near-)surface temperature or snow and ice
layers (Fettweis et al., 2013a, 2017; Wyard et al., 2020; Delhasse et al., 2020; Hanna et al., 2020; Amory et al., 2021; Kittel et al., 2022), knowing that the general circulation is forced at its lateral boundaries.

To keep the usual MAR pressure description while simulating the radiative effects of the stratosphere, additional pressure layers are added on top of the MAR grid just before calling ecRad and upon preparing the greenhouse gas and aerosol forcings as described in Sect. 3.1. This means the MAR model can run with its usual vertical discretization while still receiving radiative
fluxes from ecRad that took account of properties of the stratosphere such as the peak ozone concentration. This also means, however, that some MAR variables fed to ecRad must be set or inferred with respect to the additional stratospheric layers so that they are consistent with real world observations.

Based on the small water content in the stratosphere and previous work by Hogan and Matricardi (2020), the values of the additional stratospheric layers for the input variables required by ecRad were set as follows. First, it assumed that there is no
cloud, no liquid water, no ice crystals and no form of precipitation. Second, the top pressure layer of the extended MAR grid must include the stratopause (1 hPa), and specific humidity in all stratospheric layers is set to $3.1 \times 10^{-6}$ kg kg$^{-1}$, i.e. the typical median value across the stratosphere according to Hogan and Matricardi (2020), while the vapor saturation threshold is set to the value for 0°C, i.e. $3.8 \times 10^{-3}$ kg kg$^{-1}$, to prevent condensation and stay consistent with the previous assumptions. Finally, it is assumed the temperature at the stratopause is approximately 60 K warmer than at the tropopause (Hogan and
Matricardi, 2020). For a given air column, the temperature for the stratospheric layers can be inferred by finding the plausible tropopause in the column (which is not necessarily the top of said column), adding 60 K to obtain the temperature at the stratopause, then linearly interpolate the intermediate values.





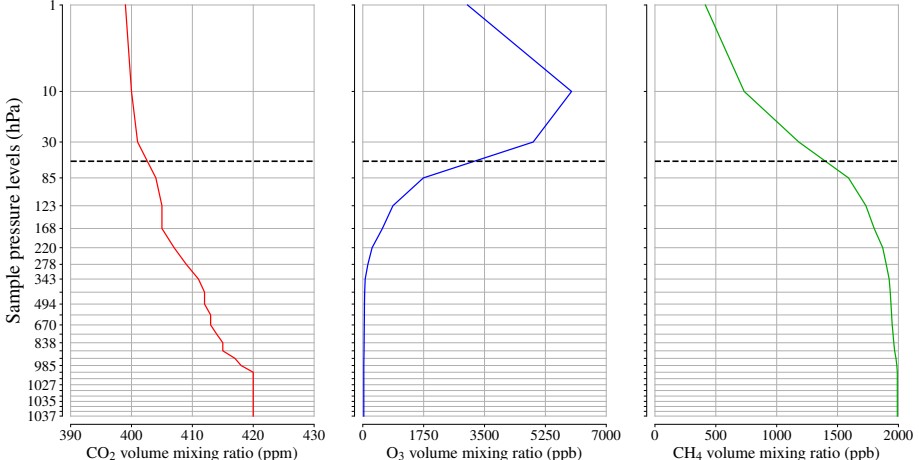

**Figure 3.** Volume mixing ratios for three greenhouse gases after fitting the forcings (from Figure 2) in the MAR grid (Sect. 3.1). Due to horizontal smoothing, the mixing ratios only vary vertically. MAR using sigma coordinates, the pressure values are based on a representative air column. The dashed line delimits the extra layers added to take account of the radiative effects of the stratosphere (Sect. 3.2).

The immediate benefit of adding stratospheric layers on top of the MAR grid is to allow ecRad to capture the radiative effects of the peak concentration of ozone, with the greenhouse gas and aerosol concentrations at the extra pressure layers being

prepared with the same method as in Sect. 3.1. The number and limits of the extra pressure layers can be freely configured by MAR users. By default, MAR running with ecRad adds three extra pressure layers on top of the MAR grid, respectively extending from 0 hPa to 5.5 hPa, from 5.5 hPa to 20 hPa and from 20 hPa to around 50 hPa, this last limit depending on the top of the MAR grid. The greenhouse gas and aerosol concentrations for each of these three layers are adjusted with the concentrations closest to 1 hPa (stratopause), 10 hPa and 30 hPa (respectively) in the forcings, yielding a total of ozone that

is close to the total of ozone from the forcings when vertically integrated, as shown in Figures 3 and 4.

### 3.3  Updated cloud fraction parameterization

Clouds are well known for playing a major role in distributing the radiative budget of the Earth (Ramanathan et al., 1989; Shupe and Intrieri, 2004), with direct consequences for surface processes. For instance, by combining climate simulations and satellite observations, Van Tricht et al. (2016) demonstrated that clouds were enhancing ice sheet meltwater runoff in

Greenland. Properly representing clouds within a climate model is therefore necessary to compute radiative fluxes that are consistent with real-world observations. When it comes to research involving the MAR model, Fettweis et al. (2017) also noted that the biases in the radiative fluxes produced by the Morcrette scheme embedded in MAR were at least partly due to an underestimated cloudiness, the radiation scheme also requiring cloud fraction values for each grid cell among its inputs.

While accurately representing clouds within a computer model is whole research topic of its own (Sundqvist et al., 1989;

Xu and Randall, 1996; Tompkins, 2002; Shonk et al., 2010; Weverberg et al., 2013), the prediction of cloudiness in the MAR model needed an update to improve the cloud fraction values it feeds to ecRad in order to further improve its output radiative




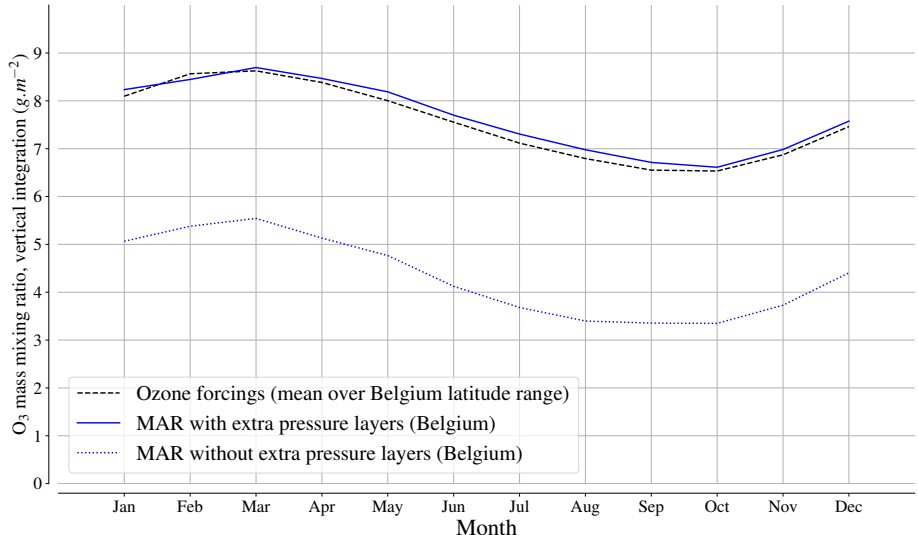

**Figure 4.** Comparison between the initial ozone volume mixing ratios and the same mixing ratios fitted to a MAR grid over Belgium, with and without the extra pressure layers (concentrations adjusted to those found at $1\,\mathrm{hPa}$, $10\,\mathrm{hPa}$ and $30\,\mathrm{hPa}$), after vertical integration ($\mathrm{g\,m^{-2}}$) and on a monthly basis, based on the outputs of a sample MAR v3.14 run over Belgium in 2019.

fluxes. Indeed, the code interfacing MAR with the Morcrette radiation scheme still computes cloud fraction values with an old formula from the ECMWF for predicting cloudiness at a large scale. Though this old approach can still be used, the cloud fraction parameterizations from Sundqvist et al. (1989) and Xu and Randall (1996) were added in the MAR code computing

the cloud fraction values sent to ecRad, with MAR users being able to choose between all three parameterizations.

The parameterizations from Sundqvist et al. (1989) and Xu and Randall (1996) were included in MAR for two reasons. First, recent work by Wang et al. (2022) re-evaluated them with CloudSat data and showed that both parameterizations struggle with predicting cloud vertical structure but are good at predicting the total cloud cover, which is currently the most important requirement for MAR regarding cloudiness, given that research involving the MAR model focuses mostly on (near-)surface

processes, as previously mentioned in Sect. 3.2. Second, both parameterizations are immediate in terms of implementation, as they only need simple variables from a given grid cell to compute a cloud fraction (CF) for that cell, without considering adjacent cells or grid-wide phenomena. On the one hand, Sundqvist et al. (1989) is solely based on relative humidity (RH), using a *critical* relative humidity threshold ($\mathrm{RH}_c$) to determine a plausible cloud fraction:

$$\mathrm{CF} = 1 - \sqrt{\frac{1 - \mathrm{RH}}{1 - \mathrm{RH}_c}} \tag{1}$$


where the critical threshold depends on the horizontal resolution of the MAR grid in kilometers ($\mathrm{dx}$) and the underlying type of surface (Wang et al., 2022):





$$
\mathrm{RH}_c = \begin{cases} 0.7 + \sqrt{\frac{1}{25+\mathrm{dx}^3}} & \text{if above land} \\ 0.81 + \sqrt{\frac{1}{50+\mathrm{dx}^3}} & \text{above sea} \end{cases} \tag{2}
$$

On the other hand, the parameterization of Xu and Randall (1996) needs relative humidity (RH), the total mixing ratio ($q_t$) of non-gaseous water species (i.e., droplets, ice crystals and snowflakes), the vapor saturation threshold ($q_s$) and 3 empirical parameters $p$, $\alpha$ and $\gamma$:

$$
\mathrm{CF} = \mathrm{RH}^p \left[ 1 - exp(-\frac{\alpha q_t}{[(1-\mathrm{RH})q_s]^\gamma}) \right] \tag{3}
$$

where $p$, $\alpha$ and $\gamma$ were empirically determined as $p = 0.25$, $\gamma = 0.49$ and $\alpha = 100$ (Xu and Randall, 1996). Due to their simplicity, the parameterizations from Sundqvist et al. (1989) and Xu and Randall (1996) were preferred over more physically realistic solutions, such as more advanced diagnostic parameterizations (Weverberg et al., 2021b, a) or prognostic solutions (Tompkins, 2002), both requiring more implementation work. The inclusion of these more complex solutions in the MAR model is left for future work.

### 3.4 Default configuration of ecRad in MAR

The ecRad radiative transfer scheme includes a large number of options (Hogan and Bozzo, 2018; Hogan, 2024a). For simplicity's sake, ecRad was configured in MAR with its default or most modern options, which can be exhaustively reviewed in the documentation from the ecRad wiki (Hogan, 2024b). For instance, MAR always enables ecRad to simulate scattering of longwave radiation by clouds, given that such a feature adds an extra computational cost of only 4% in ecRad (Hogan and Bozzo, 2018). Likewise, MAR always prepares its aerosol forcings for ecRad with a monthly aerosol climatology compliant with the CAMS aerosol specification (Flemming et al., 2017) rather than the old aerosol climatology of Tegen et al. (1997), with which MAR can also enable ecRad to simulate the longwave scattering effect of aerosols. Still for simplicity's sake, MAR re-uses the parallelization strategy of the stand-alone implementation of ecRad (Hogan, 2024b, a). It consists of processing one transect of the MAR grid at once while assigning (groups of) air columns from this slice to distinct parallel processes, the design of ecRad making this strategy very easy to implement.

Among the three available solvers in ecRad, the Tripleclouds scheme (Shonk and Hogan, 2008) was picked as the default solver to be used by ecRad in MAR. This choice constitutes a compromise between the research purpose of MAR and the overall computational cost of running ecRad. Indeed, the McICA (Pincus et al., 2003) was designed with operational weather forecasting in mind, while running ecRad with SPARTACUS (Schäfer et al., 2016; Hogan et al., 2016), currently the most advanced of the three solvers, is significantly slower than with both McICA and Tripleclouds (Hogan and Bozzo, 2018). Of





course, though Tripleclouds is the preferred radiation solver to be used by ecRad embedded in MAR, MAR users can still configure ecRad to run with any of the available solvers.

Finally, two major parameters of ecRad will be subsequently tuned to design sensitivity experiments in Sect. 4. The first is the gas-optics scheme: due to their novelty, high resolution gas-optics models built by ecCKD (Hogan and Matricardi, 2022)
needs to be tested in a regional climate model like MAR before using them as a replacement to the classical RRTM-G scheme (Mlawer et al., 1997). The second is the fractional standard deviation of in-cloud water content, denoted as $f_w$ and defined as the standard deviation of in-cloud water content divided by its mean (Shonk et al., 2010). This parameter controls the inhomogeneity of water content in clouds modeled by the solvers: the smaller $f_w$ is, the more homogeneous clouds will be. Based on multiple data sources, Shonk et al. (2010) recommends using $0.75 \pm 0.18$, with global climate models in mind. MAR
therefore uses $0.75$ by default. As MAR is a regional climate model, $f_w$ may be set with lower values, such as $0.5$.

## 4 Regional evaluation of MAR v3.14 for Belgium

### 4.1 Methodology

The ecRad radiation scheme is available in the MAR model since its version 3.14.0 (Fettweis and Grailet, 2024). For legacy reasons, but also to ease comparison between the new and the old radiation schemes, MAR v3.14 lets users decide whether to
use Morcrette or ecRad as the radiation scheme before compilation. As a result, all simulations discussed in this paper were run with MAR v3.14, with some running with the Morcrette scheme and the others with ecRad.

Our methodology is the following. MAR v3.14 has first been configured with the same study domain as Wyard et al. (2017), i.e., a grid centered on Belgium with a resolution of 5 by 5 km with 24 pressure layers in sigma coordinates extending from the surface to the low stratosphere. We then selected the 2011–2020 decade as our period of simulation both to evaluate MAR
v3.14 on an extended period and to benefit from the most recent data products of the RMIB. As these data products provide daily values, MAR v3.14 was tuned to output daily values as well (means or totals, depending on the variable). The boundary forcings for all simulations have been generated with the ERA5 dataset (Hersbach et al., 2020) for the 2011–2020 decade.

A total of nine simulations have been conducted and are listed with their respective configuration in Table 2. By running nine experiments, each with a unique configuration, we complete two tasks. First, we assess the climate sensitivity of MAR v3.14
to various configurations of ecRad and establish which configuration works best for Belgium. Second, we evaluate whether or not a well tuned MAR v3.14 can produce better radiative fluxes with ecRad than with its previous configuration (i.e. still using the Morcrette scheme) with no negative trade-off, that is, no negative impact on other MAR outputs and without tuning the output of the radiation scheme.

Our first two experiments, M1 and M2, consisted of running MAR v3.14 with the Morcrette scheme, respectively without and
with a heat fluxes tuning mechanism inherited from previous versions of MAR, which occurs right after heat fluxes have been deduced from the outputs of Morcrette. Historically, this *ad hoc* mechanism has been implemented in previous MAR versions (up to version 3.13) to sligthly mitigate known radiative fluxes biases w.r.t. ground observations that have been observed by previous research involving MAR (Fettweis et al., 2017; Wyard et al., 2018; Delhasse et al., 2020; Kittel et al., 2022).





**Table 2.** Configurations of all nine MAR (v3.14) simulations discussed in this paper, run over Belgium for the 2011–2020 decade. All experiments with ecRad use the Triplecloud radiation solver. Each simulation was given a name for readability's sake.

| Name | Radiation | Gas-optics | Extra layers | Cloud fraction | Other parameters |
|------|-----------|------------|--------------|----------------|------------------|
| M1 | Morcrette | SW: custom (6 b.), LW: RRTM-G (16 b.) | None | Initial | No heat fluxes tuning |
| M2 | Morcrette | SW: custom (6 b.), LW: RRTM-G (16 b.) | None | Initial | With heat fluxes tuning |
| E1 | ecRad | RRTM-G (bands: 14 SW, 16 LW) | None | Initial | $f_w = 0.75$ |
| E2 | ecRad | RRTM-G (bands: 14 SW, 16 LW) | 3 (0 to $\pm$ 50 hPa) | Initial | $f_w = 0.75$ |
| E3 | ecRad | RRTM-G (bands: 14 SW, 16 LW) | 3 (0 to $\pm$ 50 hPa) | Xu & Randall | $f_w = 0.75$ |
| E4 | ecRad | RRTM-G (bands: 14 SW, 16 LW) | 3 (0 to $\pm$ 50 hPa) | Sundqvist | $f_w = 0.75$ |
| E5 | ecRad | RRTM-G (bands: 14 SW, 16 LW) | 3 (0 to $\pm$ 50 hPa) | Xu & Randall | $f_w = 0.5$ |
| E6 | ecRad | RRTM-G (bands: 14 SW, 16 LW) | 3 (0 to $\pm$ 50 hPa) | Sundqvist | $f_w = 0.5$ |
| E7 | ecRad | ecCKD (bands: 44 SW, 13 LW) | 3 (0 to $\pm$ 50 hPa) | Sundqvist | $f_w = 0.75$, spectral outputs |

The seven remaining experiments all ran with the ecRad radiation scheme. E1 to E6 have been designed to evaluate not only ecRad itself but also the effects of the additions and parameters discussed in Sect. 3. The very first ecRad experiment E1 simply ran ecRad with its default parameters, briefly discussed in Sect. 3.4, and none of the adjustements described in Sect. 3. Starting from E2, stratospheric pressure layers (Sect. 3.2) are added, and the four next experiments (from E3 to E6) test the newly added cloud fraction parameterizations (Sect. 3.3) and two different values for the $f_w$ parameter of ecRad (Sect. 3.4).

The final ecRad experiment, E7, reuses the configuration of E4 but swaps the classical RRTM-G gas-optics scheme with high resolution ecCKD gas-optics models (Hogan and Matricardi, 2022) with two goals in mind. On the one hand, it is meant to verify whether or not swapping RRTM-G with more modern, high resolution gas-optics models has any negative impact on MAR outputs. On the other hand, E7 was also configured to output spectral shortwave fluxes, whose evaluation and first application, i.e. UV index prediction, are discussed in Sect. 5. For this purpose, the ecCKD models of E7 feature 44 bands in the shortwave and 13 in the longwave. To facilitate accurate calculation of UV index, the former includes 21 bands in the 280–400 nm region. To represent spectral variation of gas absorption within bands, the bands are divided further into "g-points", such that the total number of quasi-monochromatic spectral intervals is 96 in the shortwave and 64 in the longwave. This is fewer than the 112 and 140 used by RRTM-G in the shortwave and longwave, respectively, resulting in ecCKD being more computationally efficient. It was found that shortwave gas-optics models generated by ecCKD version 1.4 and earlier tended to underestimate surface spectral UV fluxes compared to benchmark line-by-line radiation calculations, which was fixed by increasing the weight of the UV fluxes in the optimization step of the Hogan and Matricardi (2022) algorithm. The shortwave ecCKD model used in this paper is from version 1.6 of ecCKD.

## 4.2 Evaluation of MAR physical variables

We assess four classical physical output variables of MAR for all nine experiments from Table 2: daily average near-surface (around 2 m above ground) temperature, daily precipitation total, daily mean surface shortwave downward fluxes and daily





mean surface longwave downward fluxes. The first three variables are compared to the gridded products provided by the
RMIB, which give daily means or daily totals (in the case of precipitation), with the daily mean temperature being based on
observations at 2 m above grass. These products cover the entire 2011–2020 decade and were built by interpolating ground
observations recorded at weather stations scattered across Belgium (Journée and Bertrand, 2010, 2011; Journée et al., 2015).
The daily mean shortwave downward fluxes have been further refined by merging the ground observations with satellite mea-
surements provided by the EUMETSAT Satellite Application Facility on Land Surface Analysis (Journée and Bertrand, 2010;
Trigo et al., 2011a). The daily mean longwave downward fluxes, i.e. the last of the four assessed MAR physical variables, are
directly compared to MSG daily `DSLF` dataset (`MDIDSLF`), a gridded product covering Europe, Africa, part of South America
and Middle-East at 0.05 degree latitude-longitude resolution which provides downward surface longwave fluxes as recorded
by the MSG satellites of EUMETSAT (Trigo et al., 2011a), due to a lack of a gridded RMIB longwave product equivalent to
the shortwave product. According to Trigo et al. (2011b), the `DSLF` product meets its target accuracy for more than 80% of its
values, that is, a relative error below 10% compared to land-based observations.

For all experiments and for all variables, the time series for each grid cell from the corresponding RMIB/MSG product
is directly compared with the time series from the MAR grid cell that is the closest in geographical coordinates, as all used
products feature a nearly identical resolution to the MAR grid. RMIB products have an equivalent resolution of 5 by 5 km
grid cells, only with a different projection system, while the MSG satellites longwave product has a resolution of 0.05 by
0.05 degree in latitude and longitude. Once MAR grid cells are paired with comparable grid cells from the data products, the
correlation, root-mean-square error (RMSE) and bias are computed for each couple of time series. The resulting 2-D statistics,
which can be visualized on the grids of the data products, have been both saved and averaged, with Table 3 providing the
average statistics.

Table 3 demonstrates, on the one hand, that all experiments performed well with respects to the RMIB products when it
comes to the daily mean temperature and the daily precipitation total: all simulations yielded an average correlation over the
Belgian territory above 0.98 for daily mean temperature and above 0.59 for daily precipitation total, with fairly low biases,
respectively significantly below 0.3 °C and 0.1 mm in absolute value. The temperature statistics are marginally better with
ecRad simulations, with lower mean root-mean-square errors and slightly higher mean correlations. The average statistics for
radiative fluxes, on the other hand, are more contrasted. While all simulations yielded very good average correlation values, the
shortwave fluxes biases differ significantly between `M1` and `M2` and the ecRad experiments. In particular, the average shortwave
fluxes biases are steadily decreasing from `E1` to `E4`. The differences with respects to the MSG longwave product are likewise
steadily decreasing from `E1` to `E4`, with `E4` and `E7` providing the most balanced radiative fluxes, their mean decadal biases for
shortwave radiation and their mean differences w.r.t. MSG for longwave radiation being both in the $[-2, 2]$ W m$^{-2}$ range.

Moreover, the ecRad experiments demonstrate the benefits of the additional pressure layers (cf. Sect. 3.2) and the new
cloudiness parameterizations (cf. Sect. 3.3 and Sect. 3.4). First, `E1` yielded the largest bias for shortwave radiation and the
largest difference for longwave radiation, due to none of the adjustments from Sect. 3 being used. Starting from `E2`, three
stratospheric pressure layers are always added on top of the MAR grid (using the configuration given in Sec. 3.2), cutting down
the average shortwave bias by almost 1.5 W m$^{-2}$ in the process. By respectively using the cloud fraction parameterizations of





**Table 3.** Evaluation statistics for the main MAR output variables for all experiments from Table 2 and for the 2011–2020 period. All variables, from both MAR and the data products, are (near-)surface daily averages (daily total for precipitation). All statistics are 2-D averages of gridded statistics computed by comparing times series from the reference data (longwave from EUMETSAT MSG satellites, the rest from RMIB) and from the MAR grid points that are the closest in geographical coordinates.

| Radiation scheme | | Morcrette | | ecRad | | | | | | |
|---|---|---|---|---|---|---|---|---|---|---|
| **MAR simulation** | | **M1** | **M2** | **E1** | **E2** | **E3** | **E4** | **E5** | **E6** | **E7** |
| Temperature (C) | Correlation | 0.980 | 0.980 | 0.981 | 0.981 | 0.981 | 0.981 | 0.981 | 0.981 | 0.981 |
| | RMSE | 1.37 | 1.35 | 1.38 | 1.37 | 1.35 | 1.35 | 1.34 | 1.34 | 1.34 |
| | Bias | -0.01 | 0.00 | 0.24 | 0.21 | 0.20 | 0.22 | 0.16 | 0.18 | 0.21 |
| Precipitation (mm) | Correlation | 0.593 | 0.593 | 0.590 | 0.591 | 0.593 | 0.592 | 0.592 | 0.593 | 0.592 |
| | RMSE | 3.66 | 3.67 | 3.69 | 3.68 | 3.67 | 3.67 | 3.67 | 3.67 | 3.67 |
| | Bias | -0.08 | -0.08 | -0.06 | -0.06 | -0.06 | -0.06 | -0.07 | -0.07 | -0.07 |
| Shortwave ($W\,m^{-2}$) | Correlation | 0.940 | 0.939 | 0.945 | 0.945 | 0.944 | 0.947 | 0.941 | 0.943 | 0.947 |
| | RMSE | 35.66 | 34.04 | 35.18 | 34.38 | 32.95 | 31.14 | 33.43 | 32.17 | 31.06 |
| | Bias | 2.64 | -4.95 | 12.42 | 11.03 | 4.57 | 0.67 | 1.04 | -2.71 | 0.36 |
| Longwave ($W\,m^{-2}$) | Correlation | 0.851 | 0.859 | 0.876 | 0.876 | 0.871 | 0.884 | 0.864 | 0.877 | 0.882 |
| | RMSE | 20.92 | 19.60 | 20.23 | 20.15 | 19.19 | 17.98 | 19.47 | 18.37 | 18.10 |
| | Difference | -5.71 | 0.00 | -8.20 | -8.07 | -3.85 | -1.50 | -2.82 | -0.43 | -1.77 |

Xu and Randall (1996) and Sundqvist et al. (1989), `E3` and `E4` significantly reduce both their bias for shortwave radiation and their difference w.r.t. MSG satellites for longwave radiation with respects to `E1` and `E2` while maintaining similar correlation and providing lower root-mean-squared errors. In particular, `E4` brings down the mean bias for shortwave fluxes to near zero, while reducing the mean difference w.r.t. MSG satellites for longwave fluxes to only -1.5 $W\,m^{-2}$.

`E5` and `E6`, also respectively using the parameterizations of Xu and Randall (1996) and Sundqvist et al. (1989) to compute cloud fraction values, further enhance cloudiness with respects to `E3` and `E4` by changing the value of the $f_w$ parameter in ecRad, which controls the homogeneity of the in-cloud water content (cf. Sect. 3.4). The default value of $0.75$ suggested by Shonk et al. (2010) is replaced for both simulations by $0.5$, a lower value which falls slightly outside the recommended range of $0.75 \pm 0.18$ but coincides with the lower mean $f_w$ values discussed by Shonk et al. (2010), which were derived from high resolution datasets. By lowering $f_w$ to 0.5 with respects to `E3`, `E5` yielded a mean shortwave fluxes bias only slightly worse than `E4`, but at the cost of worsening both the mean correlation and the mean RMSE. Likewise, `E6` worsens the mean correlation and RMSE with respects to `E4`, though it brings the mean difference for longwave fluxes close to zero while turning the mean bias for shortwave fluxes negative. However, all three `E4`, `E5` and `E6` provides better mean correlation and RMSE than both `M1` and `M2` while providing a better radiative balance on a decadal basis. Finally, `E7` provides comparable results to `E4`, which means swapping the old RRTM-G scheme with ecCKD gas-optics schemes for both shortwave and longwave radiation has no

 

**Table 4.** Statistics for the shortwave downward radiative fluxes (daily means) per season for all experiments from Table 2. Again, statistics are 2-D averages of gridded statistics much like in Table 3, but the covered periods are adjusted to the four seasons. Underlined statistics correspond to significant biases (i.e., larger than the average seasonal daily standard deviation of the RMIB data).

| Radiation scheme | | Morcrette | | ecRad | | | | | | |
|---|---|---|---|---|---|---|---|---|---|---|
| MAR simulation | | M1 | M2 | E1 | E2 | E3 | E4 | E5 | E6 | E7 |
| Winter (DJF) | Correlation | 0.830 | 0.830 | 0.857 | 0.858 | 0.853 | 0.859 | 0.844 | 0.850 | 0.859 |
| | RMSE | 18.33 | 19.15 | 15.72 | 15.61 | 15.88 | 15.58 | 16.68 | 16.42 | 15.56 |
| | Bias | -7.40 | -9.86 | 0.29 | -0.27 | -2.50 | -3.08 | -4.19 | -4.76 | -3.24 |
| Spring (MAM) | Correlation | 0.894 | 0.893 | 0.898 | 0.898 | 0.897 | 0.902 | 0.894 | 0.898 | 0.902 |
| | RMSE | 40.77 | 40.21 | 38.71 | 38.15 | 37.77 | 36.40 | 38.63 | 37.76 | 36.39 |
| | Bias | 1.41 | -7.84 | 12.16 | 10.57 | 4.83 | 0.80 | 0.51 | -3.38 | 0.54 |
| Summer (JJA) | Correlation | 0.844 | 0.843 | 0.843 | 0.844 | 0.847 | 0.861 | 0.845 | 0.859 | 0.862 |
| | RMSE | 48.12 | 43.58 | 50.20 | 48.70 | 44.94 | 41.27 | 44.67 | 41.99 | 41.04 |
| | Bias | 18.78 | 6.00 | 29.74 | 27.50 | 17.20 | 8.58 | 12.15 | 3.84 | 8.14 |
| Autumn (SON) | Correlation | 0.905 | 0.904 | 0.917 | 0.917 | 0.910 | 0.914 | 0.904 | 0.908 | 0.913 |
| | RMSE | 27.23 | 26.90 | 25.62 | 25.06 | 24.86 | 24.17 | 26.05 | 25.59 | 24.19 |
| | Bias | -2.35 | -8.16 | 7.29 | 6.14 | -1.33 | -3.64 | -4.33 | -6.56 | -4.05 |

negative impact on MAR outputs when it comes to a central Europe region such as Belgium, with only a marginal improvement for shortwave fluxes and equally marginally worse statistics for longwave fluxes.

To extend the analysis of the improved radiative fluxes obtained with ecRad, Table 4 provide seasonal statistics for the daily mean shortwave radiative fluxes for all nine experiments. The statistics are produced in the exact same manner as in Table 3, except that time series are each time truncated to only provide values for 3 specific months corresponding to each of the 4 
seasons, e.g., December, January and February (DJF) for winter. On the one hand, these seasonal statistics follow similar trends to those observed in Table 3, with the mean correlations and root-mean-square errors being almost always better with ecRad, and especially starting from E3, with E4 and E7 providing arguably the best seasonal balance. On the other hand, the seasonal statistics for M1 and M2 show large seasonal disparities regardless of tuning the heat fluxes. In the case of M1, i.e. Morcrette with no tuning, the biases are quite good in spring and autumn but considerable in the winter and large in the 
summer, with a mean bias of almost +20 W m$^{-2}$. While M2 brings down the same mean bias to only +6 W m$^{-2}$, it is at the cost of significantly worsening the spring and autumn biases and making the mean winter bias statistically significant. While the two best ecRad simulations (E4 and E7) both have a slightly worse summer bias than M2, at around +8 W m$^{-2}$, they provide better statistics for all other seasons in addition to better mean correlations and root-mean-square errors for the summer. In particular, all seasonal mean biases are below 5 W m$^{-2}$ in absolute value with the exception of the summer. In conclusion, 
despite the Morcrette simulations having reasonable mean decadal biases for radiative fluxes in Table 3, the ecRad simulations exhibit a better seasonal behaviour.




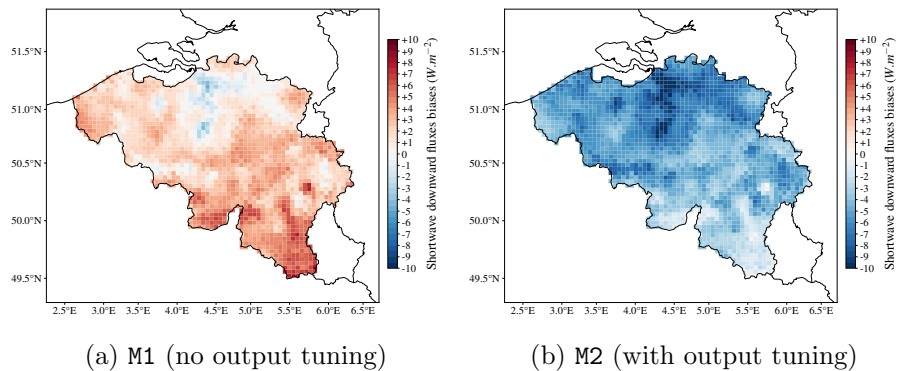

(a) `M1` (no output tuning)    (b) `M2` (with output tuning)

**Figure 5.** Shortwave downward flux biases of `M1` and `M2` (MAR v3.14 with Morcrette) for 2011–2020.

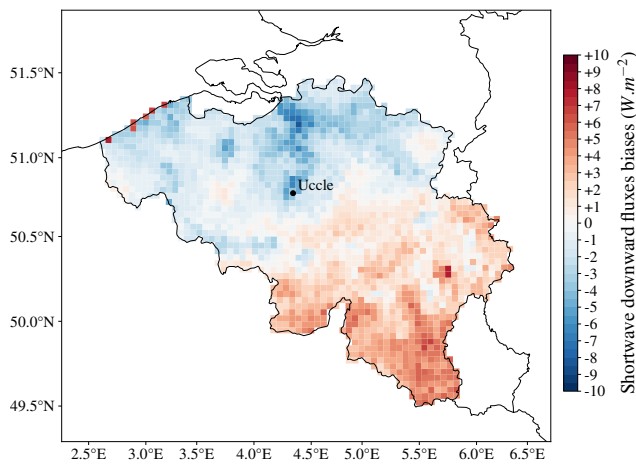

**Figure 6.** Shortwave downward flux biases of `E7` (MAR v3.14 with ecRad and ecCKD) for 2011–2020. Uccle location is given for Sect. 5.

Finally, Figures 5 and 6 illustrate the spatial variability of the shortwave fluxes biases over Belgium during 2011–2020 for three of our nine simulations: `M1` and `M2` (Fig. 5), and `E7` (Fig. 6), respectively. Figure 5 further demonstrate that the old MAR configuration struggles to provide balanced shortwave biases, regardless of tuning the heat fluxes, featuring an overwhelming

majority of positive biases without tuning the output fluxes and the opposite result with tuning. Figure 6, on the other hand, shows that MAR v3.14 running with the best configuration of ecRad offers low biases for most of the Belgian territory with a good balance between positive and negative biases. Only the southern tip of Belgium exhibits larger positive biases with `E7`, though all three maps shown in Fig. 5 and Fig. 6 exhibit their highest biases in that area.

### 4.3 Impact of ecRad on MAR computational performance

To evaluate the cost of all code changes in MAR presented in this paper, the impact on execution time of running MAR with the ecRad radiation scheme instead of the Morcrette scheme should also be assessed. To do so, the MAR model has been run





**Table 5.** Relative increases in time elasped while calling ecRad with various configurations w.r.t. the Morcrette scheme, both in MAR v3.14. Based on a single day of simulation (24 calls of radiation scheme) and using an increasing number of CPUs, with 5 repetitions.

| Relative change with | 4 CPUs | | 8 CPUs | | 12 CPUs | | Average | |
|---|---|---|---|---|---|---|---|---|
| Timed element | ecRad | MAR | ecRad | MAR | ecRad | MAR | ecRad | MAR |
| ecRad, RRTM-G, no stratospheric layers | +32.96% | +2.05% | +45.88% | -2.64% | +47.93% | -1.92% | +42.25% | -0.83% |
| ecRad, RRTM-G, stratospheric layers | +45.40% | -0.28% | +69.90% | +5.49% | +66.56% | +5.50% | +60.62% | +5.35% |
| ecRad, ecCKD, stratospheric layers | +11.48% | +1.45% | +30.88% | -1.46% | +36.33% | -0.80% | +26.23% | -0.27% |

several times for a whole day with the Belgium grid (cf. Sect. 4.1) on the same machine. Four configurations have been tested: the first is MAR v3.14 with the Morcrette scheme and the heat fluxes tuning (assumed to have a negligible computational cost), equivalent to `M2` in Table 2, while the three other configurations all use ecRad with the configuration changes that are the most likely to impact performance. The first ecRad configuration is equivalent to `E1`, i.e., ecRad embedded in MAR without the adjustments of Sect. 3 and RRTM-G as the gas-optics scheme. The second ecRad configuration corresponds to `E2` and adds the three extra pressure layers on top of the MAR grid before calling ecRad to approximate the stratosphere (cf. Sect. 3.2), therefore slightly increasing the grid size. Finally, the last configuration assessed for performance corresponds to `E7`, i.e., MAR v3.14 with ecRad, the additional pressure layers and the high resolution ecCKD gas-optics schemes. Indeed, changing the gas-optics schemes used by ecRad has a non-negligible impact on performance as ecCKD schemes are meant to be computationally efficient. All four configurations have been run on the exact same machine with an increasing number of CPUs (4, 8 then 12 CPUs). At each MAR run, the overall execution time was measured with the `time` command of Linux while the calls to the radiation scheme were individually timed with the help of the `system_clock` native Fortran function. To mitigate the randomness of the experiments, notably induced by the varying I/O cost of reading files on disk, each scenario, defined by a configuration and a number of CPUs, has been repeated a total of five times. All recorded times (overall MAR time and ecRad calls) were then averaged for each scenario. Table 5 provides the relative changes in execution time between the ecRad configurations and the Morcrette configuration, for each number of CPUs used and on average.

Table 5 shows that running MAR v3.14 with ecRad rather than Morcrette has barely an impact on the overall execution time, the difference ranging from -1.92 to +5.5%, despite the time elasped during ecRad calls being up to +70% longer compared to Morcrette calls. In particular, the average relative change in total time turned to be negative for two of the three ecRad configurations during our tests, suggesting the extra cost of running ecRad rather than Morcrette is negligible enough to be compensated by the random variation in the execution time of MAR as a whole. ecRad calls lasting longer than Morcrette calls may be attributed to the increased spectral resolution (Morcrette uses only 6 bands for shortwave radiation, cf. Sect. 2.1) as well as to the additional physical processes considered by ecRad, such as the longwave scattering of clouds and aerosols, which is always enabled in MAR (cf. Sect. 3.4).

Table 5 also shows there are non-negligible differences for radiation calls across the three ecRad configurations. Unsurprisingly, the largest increase occurred after adding the pressure layers approximating the stratosphere. By contrast, after swapping the RRTM-G scheme with ecCKD gas-optics schemes, ecRad calls became only +26% longer on average than Morcrette calls,





even with the extra pressure layers and the additional simulated physical processes. The performance of ecRad calls may be further improved by using ecCKD gas-optics schemes featuring less spectral intervals than the schemes used for this research, i.e., the current shortwave gas-optics scheme, featuring 96 spectral intervals, may be replaced with a comparable scheme with 64 spectral intervals for improved performance. In conclusion, running MAR v3.14 with the ecRad radiation scheme has practically no extra cost, especially when ecCKD gas-optics schemes are preferred over RRTM-G.

## 5 Assessment of spectral shortwave fluxes

### 5.1 Methodology

From the additional functionality within ecRad and ecCKD gas-optics models, MAR v3.14 is capable of producing fine surface spectral shortwave downward fluxes with user-defined spectral bands, provided the ecCKD algorithm has pre-computed a gas-optics model with a resolution high enough to accommodate the user's bands. As shortly discussed in Sect. 2.2, such a feature offers new research opportunities for the MAR model, and we hereby demonstrate its potential by both evaluating spectral shortwave fluxes produced by MAR v3.14 and using the same fluxes for predicting UV indices.

The UV index is a simple metric designed to inform the public about how much harmful ultraviolet radiation reaches the Earth's surface at a given time, as high doses of ultraviolet radiation at specific wavelengths can damage the human skin (WHO, 2002). UV indices below 6 correspond to low to moderate risks, while the 6–7 and 8–10 ranges correspond respectively to high and very high risks, with 11 and more, though rare, amounting to extreme danger (WHO, 2002). UV indices are typically obtained by integrating surface downward ultraviolet radiative fluxes in $W\,m^{-2}$ on the 250–400 nm spectral range while weighting them by the action spectrum for erythema, i.e. a redness of the skin that can be induced by solar radiation, as defined by ISO/CIE (1999). In particular, this spectrum gives more weight to the 250–328 nm range. Given the CIE action spectrum $s_{er}(\lambda)$, which returns a weight given a wavelength $\lambda$ expressed in nm, the UV index $I_{UV}$ is defined as

$$I_{UV} = 40 \times \int\limits_{250}^{400} s_{er}(\lambda) f_{sw}(\lambda)\,d\lambda \tag{4}$$

where $f_{sw}(\lambda)$ denotes the surface downward shortwave radiative flux in $W\,m^{-2}$ at a given wavelength $\lambda$ in nm and where the product $s_{er}(\lambda) f_{SW}(\lambda)$ is also called the erythemal irradiance (McKenzie et al., 2014).

To both evaluate the spectral fluxes produced by MAR v3.14 and use them to predict UV indices, the BISA provided us with spectral measurements captured by a spectrometer at Uccle observatory (50.797 °N, 4.357 °E, cf. Fig. 6) from late June 2017 to December 2020. These measurements, given in $mW\,m^{-2}\,nm^{-1}$, cover the 280–500 nm spectral range with a step of 0.5 nm and have been captured every 15 minutes during daytime. They are not continuous across the covered period, as they have been sporadically interrupted, and as a few days had to be omitted due to calibration issues. The 280–500 nm range covers both the UV-A and UV-B ranges (Tobiska and Nusinov, 2006), and most of the range covered by the CIE action spectrum




for erythema, i.e. 250–400 nm (ISO/CIE, 1999); note that virtually no radiation in the range 250–280 nm penetrates to the
surface due to being completely absorbed by ozone (Hogan and Matricardi, 2020). The spectrometer also covers a third of the
Photosynthetically Active Range (PAR), defined as 400–700 nm.

To compare MAR v3.14 with the spectral observations from Uccle, the simulation E7 from Table 2 (Sect. 4.1), running
ecRad with high resolution ecCKD gas-optics models, has been configured to also produce hourly spectral shortwave fluxes in
$W\,m^{-2}$. ecRad maps the surface spectral fluxes from internal shortwave bands (determined by the ecCKD gas-optics model
in use) to user-specified output bands, assuming that the optical properties of the atmosphere are constant across each internal
shortwave band. Therefore, the spectral distribution of radiation within each band is proportional to the incoming solar spectrum
at top of atmosphere. This means that if the user requests much finer spectral output than is resolved internally, the spectral
outputs may not be accurate, even though they will integrate to the same broadband shortwave flux.

In our case, the output spectral bands we defined in our E7 simulation are virtually the same as the internal spectral bands
used in the 96 *g-points* ecCKD model for shortwave radiation used by E7 (cf. Sect. 4.1), and therefore no significant error
is incurred by the mapping. The spectral bands of E7 cover the same spectral range as the Uccle data, i.e. 280–500 nm, and
consist of 14 consecutive bands of 5 nm covering the 280–350 nm range and 15 consecutive bands of 10 nm for the 350–
500 nm range. The 14 first bands cover each a smaller range to ensure the UV-B range and the lower part of the UV-A range
are well captured enough for UV index prediction, with the part of the CIE action spectrum having the most impact on UV
indices ranging approximately from 280 to 310 nm (McKenzie et al., 2014).

## 5.2 Evaluation of spectral fluxes at Uccle

To evaluate the spectral shortwave fluxes of the E7 MAR v3.14 simulation, we post-process the BISA spectral data from Uccle
observatory into a format suitable for direct comparison. This post-processing is done in two steps. The first step consists of
numerically integrating the BISA data on the spectral bands we defined on the 280–500 nm range in E7. Given $U_{sw}(\lambda)$, a
raw spectral observation from Uccle at the $\lambda$ wavelength (by steps of 0.5 nm) and given in $mW\,m^{-2}\,nm^{-1}$, the numerical
integration $I_{\lambda_{min},\lambda_{max}}$ on the MAR spectral band $\lambda_{min}$–$\lambda_{max}$ nm in $W\,m^{-2}$ is given by

$$I_{\lambda_{min},\lambda_{max}} = \sum_{i=0}^{N} \frac{U_{sw}(\lambda_{min} + 0.5 \times i)}{1000} \times 0.5 \tag{5}$$

where $N$ is $(\lambda_{max}-\lambda_{min})\times 2-1$. Once all Uccle measurements have been numerically integrated on our 29 spectral bands, the
second step simply consists of aggregating the resulting spectral bands for a given date and hourly slot and compute the hourly
average flux per band in $W\,m^{-2}$. Doing so, the post-processed Uccle data has both the same temporal and spectral resolution
as the MAR spectral fluxes. Finally, it should be noted that, upon calling a radiation scheme for a given hour, MAR prepares
the cosine of the solar zenith angle at the half-hour to get representative fluxes for the hourly slot. Therefore, to guarantee
sunrise and dusk happen at the same time in both datasets, the Uccle times were shifted by half an hour just before computing
the hourly average spectral fluxes.





**Table 6.** Evaluation statistics of MAR spectral shortwave downward fluxes for each spectral band defined in Sect. 5.1, compared to daytime measurements recorded at Uccle between late 2017 and December 2020 that were numerically integrated on the same spectral bands. The *Mean Uccle* columns provide the mean and standard deviation of the latter. Only common dates and hours are compared.

| Band | Mean Uccle | Corr. | RMSE | Bias | Band | Mean Uccle | Corr. | RMSE | Bias |
|------|-----------|-------|------|------|------|-----------|-------|------|------|
| nm | W m$^{-2}$ | / | W m$^{-2}$ | W m$^{-2}$ | nm | W m$^{-2}$ | / | W m$^{-2}$ | W m$^{-2}$ |
| 280–285 | 7.2e−8 ± 1.3e−7 | -0.02 | 1.5e−7 | −7.2e−8 | 360–370 | 1.46 ± 1.67 | 0.91 | 0.80 | +0.16 |
| 285–290 | 5.5e−8 ± 1.2e−7 | 0.00 | 2.8e−7 | +1.9e−8 | 370–380 | 1.52 ± 1.74 | 0.91 | 0.86 | +0.20 |
| 290–295 | 7.6e−6 ± 2.4e−5 | 0.83 | 1.7e−5 | −4.3e−6 | 380–390 | 1.39 ± 1.60 | 0.91 | 0.79 | +0.17 |
| 295–300 | 5.5e−4 ± 1.3e−3 | 0.89 | 6.7e−4 | −1.7e−4 | 390–400 | 1.68 ± 1.93 | 0.91 | 0.90 | +0.14 |
| 300–305 | 0.01 ± 0.02 | 0.92 | 8.6e−3 | −2.4e−3 | 400–410 | 2.49 ± 2.87 | 0.91 | 1.48 | +0.38 |
| 305–310 | 0.05 ± 0.08 | 0.93 | 0.03 | −6.5e−4 | 410–420 | 2.67 ± 3.07 | 0.91 | 1.43 | +0.16 |
| 310–315 | 0.14 ± 0.19 | 0.93 | 0.07 | −1.3e−3 | 420–430 | 2.53 ± 2.91 | 0.90 | 1.46 | +0.31 |
| 315–320 | 0.22 ± 0.29 | 0.93 | 0.12 | +0.01 | 430–440 | 2.60 ± 2.99 | 0.90 | 1.45 | +0.23 |
| 320–325 | 0.32 ± 0.39 | 0.93 | 0.17 | +0.03 | 440–450 | 3.04 ± 3.51 | 0.90 | 1.71 | +0.28 |
| 325–330 | 0.52 ± 0.61 | 0.92 | 0.25 | +0.01 | 450–460 | 3.26 ± 3.75 | 0.90 | 1.89 | +0.35 |
| 330–335 | 0.53 ± 0.61 | 0.92 | 0.28 | +0.06 | 460–470 | 3.30 ± 3.79 | 0.90 | 1.86 | +0.28 |
| 335–340 | 0.54 ± 0.62 | 0.92 | 0.29 | +0.07 | 470–480 | 3.32 ± 3.84 | 0.90 | 1.92 | +0.32 |
| 340–345 | 0.58 ± 0.66 | 0.92 | 0.32 | +0.08 | 480–490 | 3.23 ± 3.74 | 0.90 | 1.84 | +0.24 |
| 345–350 | 0.58 ± 0.67 | 0.92 | 0.32 | +0.07 | 490–500 | 3.09 ± 3.56 | 0.90 | 1.88 | +0.40 |
| 350–360 | 1.24 ± 1.42 | 0.92 | 0.68 | +0.16 | **280–500** | **40.31 ± 46.29** | **0.91** | **22.29** | **+4.09** |

We then compare the time series of spectral shortwave downward fluxes (produced by E7) from the MAR grid cell encompassing the geographical coordinates of Uccle with the post-processed Uccle data. Since the latter is not a completely continuous time series and consists exclusively of diurnal measurements, the former has been truncated to only feature common hours. As a consequence, nocturnal time steps from MAR are omitted from our evaluation. Table 6 provides, for each

spectral band, the mean and standard deviation of Uccle fluxes in W m$^{-2}$ followed by the correlation, root-mean-square error (RMSE) and bias of the MAR spectral fluxes for the same band. To visually compare our spectral fluxes to the post-processed Uccle data, Figure 7 plots the mean and standard deviation for each MAR spectral band for both the post-processed Uccle spectral observations and MAR spectral fluxes, in function of the wavelength and re-expressed in mW m$^{-2}$ nm$^{-1}$ to ensure the shapes of the averages curves match with the magnitudes of the solar radiation reaching Earth's surface.

Table 6 and Fig. 7 both demonstrate a strong correlation between the daytime observations from Uccle and the (daytime) spectral outputs of our E7 simulation. Only the first two spectral bands have a near-zero correlation, but this can be attributed to the extremely low values recorded by the spectrometer between 280 and 290 nm. As such, they may be considered as noise rather than proper measurements. Starting from 290 nm, MAR spectral fluxes start to correlate well with the post-processed Uccle observations, the correlation rising to 0.93 at the end of the UV-B range, which ends at 315 nm (Tobiska and Nusinov,

2006). Starting from 315 and until 500 nm, the correlations remain strong but the biases and root-mean-square errors rise along





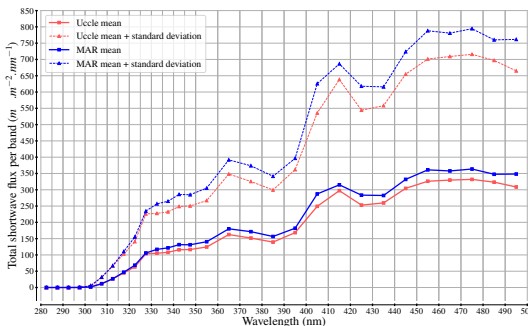

**Figure 7.** Mean and standard deviation of the spectral shortwave fluxes derived from Uccle observations (June 2017 to December 2020, day-time only) and from MAR v3.14 (E7 from Table 2). The X-axis gives the wavelengths while the Y-axis gives the fluxes in $\mathrm{mW\,m^{-2}\,nm^{-1}}$.

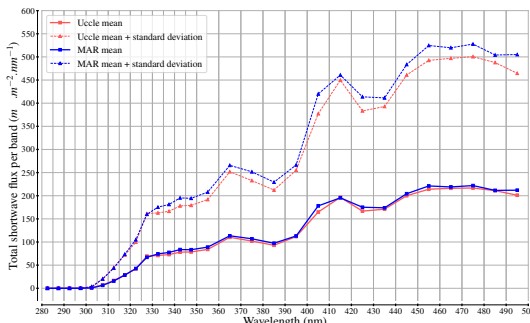

**Figure 8.** Mean and standard deviation of the spectral shortwave fluxes derived from Uccle observations (June 2017 to December 2020) and from MAR v3.14 (E7 from Table 2) after filtering the data to only keep cloudy days (daily mean cloud cover above 0.7) according to MAR.

the mean Uccle spectral fluxes. Figure 7 highlights this very well, as it simultaneously depicts a nearly perfect match between the MAR spectral fluxes and the post-processed Uccle fluxes for the UV-B range (280–315 nm) and consistently positive biases of MAR spectral fluxes for the rest of the spectrum, though these biases always stay below $50\,\mathrm{mW\,m^{-2}\,nm^{-1}}$ (or $0.5\,\mathrm{W\,m^{-2}}$ for a spectral band of 10 nm).

The consistent biases of our E7 simulation in the 315–500 nm range may be explained by an underestimated cloudiness in MAR. On the one hand, post-processed Uccle observations may be impacted by cloud cover variations occurring within a same hourly slot, while the MAR model predicts cloud fraction values for a whole hourly slot before calling the radiation scheme. On the other hand, as evidenced by our evaluation of physical variables in Sect. 4.2, there remain seasonal biases for shortwave downward fluxes, even with our best configurations. In particular, large positive biases were obtained during the summer by all

of our simulations, even while comparing daily averages of shortwave fluxes, which may be due to underestimated cloudiness during this season. One way to test this cloudiness hypothesis consists of filtering our time series of spectral fluxes to only keep





the time steps from cloudy days according to our E7 simulation, i.e., when the daily mean cloud cover is above 0.7. Figure 8 plots the mean and standard deviation for each spectral band in the 280–500 nm range in the same manner as Fig. 7, but only using cloudy days according to MAR. Naturally, this changes the magnitude of the average fluxes, the highest means dropping

from around 350 mW m$^{-2}$ nm$^{-1}$ to only slightly above 220 mW m$^{-2}$ nm$^{-1}$. However, in these conditions, the agreement between the MAR spectral fluxes and the post-processed Uccle measurements improves, the largest biases dropping below 25 mW m$^{-2}$ nm$^{-1}$ with the differences in standard deviation dropping below 50 mW m$^{-2}$ nm$^{-1}$. This suggests the biases in Fig. 7 may be due to a non-negligible number of time steps where the cloud clover was weaker in MAR than it was in reality.

### 5.3 A first application: UV index prediction

The very good agreement between the spectral shortwave fluxes of our E7 simulation and the Uccle spectral observations in the UV-B range (i.e., 280–315 nm) makes MAR v3.14 running with ecRad a credible candidate for predicting UV indices. Therefore, we hereby apply the concept of UV index to both the Uccle spectral observations and the shortwave spectral fluxes from E7. While the BISA did not provide us with UV index data from Uccle over the same period as the spectral observations, the high resolution of these observations, which measured shortwave radiative fluxes per steps of 0.5 nm, should lead to

realistic UV indices. We can therefore compare the UV indices derived from the observations to the indices predicted on the basis of MAR spectral fluxes to assess whether or not MAR v3.14 can predict credible UV indices while defining a few dozens of spectral bands, assuming these spectral bands are not finer than the spectral resolution of the ecCKD gas-optics model for shortwave radiation used internally by ecRad.

As explained in Sect. 5.1 and formulated in (4), the UV index is essentially an integration of surface ultraviolet radiative
fluxes in W m$^{-2}$ on the 250–400 nm spectral range weighted by the CIE action spectrum (ISO/CIE, 1999). The CIE action spectrum for erythema $s_{er}(\lambda)$ is formally defined as

$$s_{er}(\lambda) = \begin{cases} 1.0 & \text{if } \lambda \in [250, 298] \text{ nm} \\ 10^{0.094(298-\lambda)} & \text{if } \lambda \in ]298, 328] \text{ nm} \\ 10^{0.015(140-\lambda)} & \text{if } \lambda \in ]328, 400] \text{ nm} \end{cases} \tag{6}$$

where $\lambda$ is a wavelength expressed in nm in the 250–400 nm spectral range (ISO/CIE, 1999; McKenzie et al., 2014). Although
both the Uccle measurements and the spectral bands we configured in the E7 simulation begin at 280 nm, the 250–280 nm spectral range is expected to have little impact on UV indices due to radiation from the UV-C spectral range (100–280 nm) being completely absorbed by the ozone layer (Tobiska and Nusinov, 2006; Hogan and Matricardi, 2020). In other words, the most relevant ranges for UV index prediction are the UV-B range and the beginning of the UV-A range, on which E7 is in very good agreement with the numerically integrated Uccle observations (cf. Sect. 5.2).

Since both the Uccle spectral observations and the MAR spectral fluxes are provided in small, fine spectral bands, though the former has a higher resolution than the latter, we can predict UV indices by performing the numerical equivalent of the





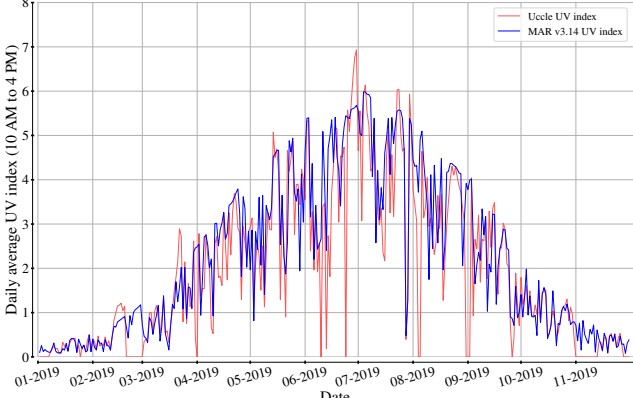

**Figure 9.** Daily average UV index (based on hourly indices between 10 AM and 4 PM) at Uccle based respectively on the BISA data and the E7 spectral shortwave fluxes (MAR v3.14) during the year 2019 (most complete year in BISA data).

continuous integration (4). In other words, we will compute UV indices as weighted sums of our spectral fluxes, the weights being defined by the CIE action spectrum formally defined by (6).

The UV indices derived from Uccle observations are calculated as follows. First, we numerically integrate the raw measure-
ments between 280 and 400 nm on bands of 1 nm in order to eliminate the factor 2 between the initial measurements (by steps of 0.5 nm) and their unit ($\mathrm{mW\,m^{-2}\,nm^{-1}}$) before averaging the fluxes on an hourly basis, again to match with the temporal resolution of MAR. Then, the UV indices based on Uccle data are obtained by computing the numerical equivalent of (4). The UV indices based on the spectral shortwave fluxes produced by our E7 simulation are calculated mostly in a similar manner, except that we compute for each MAR spectral band an average CIE action spectrum weight $W_{\lambda_{min}}^{\lambda_{max}}$ as

$$W_{\lambda_{min}}^{\lambda_{max}} = \frac{1}{\lambda_{max} - \lambda_{min}} \times \sum_{\lambda=\lambda_{min}}^{\lambda_{max}-1} \mathrm{s_{er}}(\lambda) \tag{7}$$

where $\lambda_{min}$ and $\lambda_{max}$ denote respectively the lower and upper bounds of a MAR spectral band defined on the $\lambda_{min}$–$\lambda_{max}$ nm range. With $W_{\lambda_{min}}^{\lambda_{max}}$ defined, UV indices based on E7 are calculated with

$$\mathrm{I_{UV}} = 40 \times \left[ \sum_{i=1}^{14} \mathrm{W}_{280+5(i-1)}^{280+5i} \mathrm{M_{sw}}(i) + \sum_{i=15}^{19} \mathrm{W}_{350+10(i-15)}^{350+10(i-14)} \mathrm{M_{sw}}(i) \right] \tag{8}$$


where $M_{sw}$ denotes a spectral shortwave flux produced by our E7 simulation in $\mathrm{W\,m^{-2}}$ and where $i$ denotes one of the 19 consecutive spectral bands over 280–400 nm tuned in E7 and exhaustively enumerated in Table 6 (Sect. 5.2).

Using the numerical equivalent of (4) and the formulas presented in (7) and (8), UV indices can be computed off-line directly from the BISA data and the E7 spectral fluxes. We compare the time series of the UV indices respectively derived from the





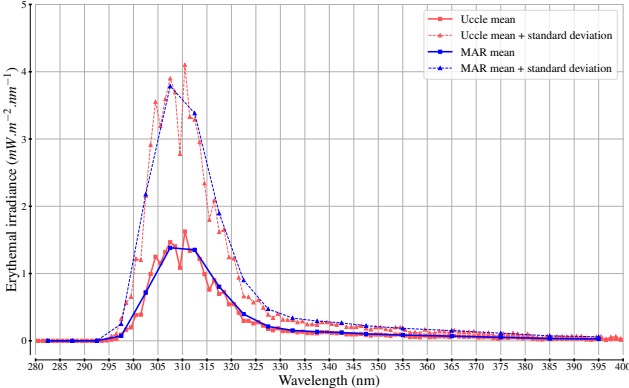

**Figure 10.** Mean and standard deviation for erythemal irradiance (i.e., shortwave radiative fluxes weighted by the CIE action spectrum for erythema (McKenzie et al., 2014)) in function of the wavelength for both Uccle data and the E7 simulation (June 2017 to December 2020).

BISA data and from the E7 shortwave spectral fluxes from the MAR grid cell encompassing Uccle in the same way as we previously did for the spectral bands, i.e., by keeping only common dates and hours and computing the correlation, root-mean-square error and bias. Doing so, we obtain a correlation of 0.929, a RMSE of 0.671 and a bias of +0.066. When it comes to the extreme values, the observations yield a maximum UV index of 9 (9.161) versus a maximum UV index of 8 (8.172) yielded by the E7 spectral outputs. In other words, UV indices based on MAR v3.14 spectral outputs match closely with the indices

derived from the BISA data, the main issue being that the former fall short of capturing the maxima of the latter. Figure 9 allows to visualize both these observations by plotting the daily average UV index (based on hourly indices between 10 AM and 4 PM) from both time series on year 2019, i.e., the most complete year in the BISA data. Indeed, the MAR curve matches quite well the observations curve, excluding the days with missing data, but misses several of its spikes.

To explore the differences between the UV indices derived from the observations and from MAR v3.14 spectral fluxes (E7),

Fig. 10 provides the mean and standard deviation for erythemal irradiance in function of the wavelength for both the Uccle data and the E7 fluxes. Contrary to Figures 7 and 8, the differences in resolution are represented, as the UV indices based on Uccle data are calculated on the basis of nanometer-wide spectral bands. While both curves match well on the 280–300 and 325–400 nm spectral ranges, non-negligible differences appear in between, with the maxima (in mean only or with standard deviation) belonging to the Uccle curve. For instance, in the middle of the 300–310 nm range, the erythemal irradiance is

noticeably higher with Uccle observations, this part of the 300–325 nm range having slightly more weight in UV index calculation than higher wavelengths in the same range.

## 5.4  Discussion

Our evaluations of both the simulated physical variables and the spectral shortwave fluxes of MAR v3.14 allow us to gauge the benefits of including the ecRad radiative transfer scheme and to assess the current limits of MAR v3.14. On the one hand,

Table 4 from Sect. 4.2 and Figures 7 and 8 show that radiative fluxes biases remain across seasons due to underestimation or





overestimation of cloudiness (depending on the season). Figure 8 notably demonstrates that under cloudy conditions (according to MAR), positive biases of MAR with respects to observations become negligible even if fluxes are processed in spectral bands. This may suggest the overall biases are due to (partly) cloudy days in the observations being clear days in MAR, though the spatial and temporal resolution of MAR may also contribute to the underestimated cloudiness, especially when compared to
observations from one specific site that were captured with higher temporal resolution.

On the other hand, as hinted by Sect. 3.1, greenhouse gas and aerosol concentrations in MAR v3.14 vary mostly on a monthly basis due to the initial forcings consisting of monthly means. In other words, daily variations are not modeled. Moreover, as pictured by Fig. 4, the current stratosphere configuration of MAR v3.14 yields a slightly overestimated total of ozone when vertically integrated. This slight overestimation and the lack of daily variability may explain why the UV indices based on our
MAR spectral fluxes are not much higher than 8, while Uccle observations occasionally yield an UV index above 9, though this does not prevent MAR v3.14 from leading to credible UV indices on average.

Possible ways to improve the radiative fluxes of MAR v3.14 and its by-products, like UV indices in this context, therefore include an improved cloud fraction prediction (notably by considering prognostic schemes, cf. Sect. 3.3) as well as modeling the daily variation of ozone and aerosol concentrations. In particular, modeling the daily variation of the total of ozone should
improve the computation of spectral fluxes in the ultraviolet range, which should in turn lead to a more accurate prediction of the peak UV indices.

## 6   Conclusion

The physical accuracy of the regional atmospheric model MAR partly relies on its underlying radiative transfer scheme, or radiation scheme, i.e., a component simulating how both shortwave and longwave radiative fluxes evolve over time, depending
on various physical variables describing the Earth's atmosphere. For about two decades, MAR ran with a late version of the Morcrette scheme, which was notably used for the ERA-40 reanalysis (Uppala et al., 2005; Morcrette et al., 2008). Several radiation schemes have succeeded the Morcrette scheme since then, leading up to ecRad, the current radiation scheme provided by the ECMWF, which is operational in the IFS since 2017 (Hogan and Bozzo, 2018). The ecRad radiation scheme distinguishes itself from past schemes by putting emphasis on modularity, having the ability to solve any of its sub-problems
(such as solving radiation equations) with interchangeable solutions. In particular, the latest version of ecRad (Hogan, 2024a) can replace the classical RRTM-G gas-optics scheme (Mlawer et al., 1997) with high resolution gas-optics schemes built by the new ecCKD tool from the ECMWF (Hogan and Matricardi, 2022). The resulting increase in spectral resolution also makes ecRad a suitable tool for producing high resolution spectral shortwave fluxes.

This paper presented a new version of MAR, version 3.14, embedding ecRad as its new radiation scheme, and assessed its
renewed accuracy. In addition to detailing the changes brought to MAR to take advantage of ecRad, we demonstrated that a properly tuned MAR v3.14 running with ecRad can produce more balanced shortwave and longwave radiative fluxes than with Morcrette, both at the scale of a whole decade and at the scale of seasons, based on gridded surface data over Belgium provided by the RMIB (Journée and Bertrand, 2010, 2011; Journée et al., 2015) and on observations made by the EUMETSAT MSG



satellites (Trigo et al., 2011a, b) for the 2011–2020 decade. Furthermore, this improvement of radiative fluxes had no negative
consequences with either the performance of MAR or its usual physical output variables, such as near-surface temperature.

Using the latest version of ecRad and our best ecRad configuration, but swapping the classical RRTM-G gas-optics scheme
with high resolution ecCKD gas-optics schemes, we produced spectral shortwave fluxes in the 280–500 nm spectral range. By
comparing our outputs to observations captured by a spectrometer at Uccle observatory in the same range during 2017–2020
and provided to us by the BISA, we assessed the ability of both ecRad and MAR to produce realistic spectral fluxes. We
demonstrated the MAR spectral fluxes were in very good agreement with Uccle measurements, particularily over the UV-B
range (280–315 nm) and the beginning of the UV-A range (315–400 nm). Such a result led us to consider predicting UV
indices with MAR outputs. Our first attempt at this task led to credible UV indices, having a correlation of 0.929 with UV
indices derived from the Uccle observations, though MAR v3.14 falls short of matching with the highest observations-based
UV indices due to its own limits when it comes to cloudiness and the temporal variability of greenhouse gases and aerosols,
and ozone in particular.

Future work to improve the radiative fluxes predicted by the MAR model will focus on improving the representation of
clouds and increasing the temporal resolution of greenhouse gas and aerosol concentrations. More broadly, future work with
MAR will take advantage of the increased spectral resolution in the shortwave range to produce new forcings for other computer
models, and in particular those requiring spectral shortwave fluxes in the photosynthetically active region.

*Author contributions.* Jean-François Grailet designed and implemented the code interfacing MAR with ecRad (version 1.5.0), ran the sim-
ulations, wrote the code for comparing their output variables to the reference data as well as a collection of scripts used for: evaluating
spectral fluxes, producing and evaluating UV indices, and generating the various figures of this manuscript. Jean-François also wrote the
initial manuscript. Robin J. Hogan is the main author of ecRad and ecCKD and provided scientific and technical support throughout the
process of including ecRad in MAR. Robin also provided his feedback on the manuscript as well as additional scientific details. Nicolas
Ghilain provided the RMIB and EUMETSAT data used to evaluate MAR v3.14, contacted the BISA to obtain the spectral shortwave data
from Uccle, and offered his support and feedback on the evaluation methodology. Xavier Fettweis provided technical support for updating
and running the MAR model and finalized the source code of MAR v3.14. Marilaure Grégoire planned and founded the research.

*Code and data availability.* The source code of MAR v3.14, embedding ecRad v1.5.0, can be downloaded on Zenodo (Fettweis and Grailet,
2024), along with its forcings and additional files (such as the ecCKD gas-optics model for shortwave radiation used in this research).

The ecRad radiative transfer scheme can also be downloaded on the ECMWF Confluence Wiki (Hogan, 2024b) as a stand-alone software
that can be run outside any climate model, and is also available on GitHub (Hogan, 2024a). The climatological data for greenhouse gases and
aerosols (CAMS specification) used by the IFS and re-used by MAR v3.14 can also be freely downloaded as NetCDF files on the ECMWF
Confluence Wiki (Hogan, 2024b) as well. A MAR repository is also available on GitLab (Fettweis, 2024).



*Competing interests.* The authors declare that they have no conflict of interest.

*Acknowledgements.* The authors would like to thank David Bolsée from the Royal Belgian Institute for Space Aeronomy (BISA), who provided us with the spectral measurements captured at Uccle. His research has been funded through the Solar Terrestrial Center of Excellence.

The present research benefited from computational resources made available on Lucia, the Tier-1 supercomputer of the Walloon Region, infrastructure funded by the Walloon Region under the grant agreement n°1910247.



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
