# Peer review of "Inclusion of the ECMWF ecRad radiation scheme (v1.5.0) in the MAR model (v3.14), regional evaluation for Belgium and assessment of surface shortwave spectral fluxes at Uccle"

_EGUsphere, 2024_

## Author Response (AR1)

**Response to Referee 1 (RC1)**

> A nice paper overall but think some improvements are needed. I like the inclusion of the UV index work. Perhaps more work is needed to show that ecRad is as good if not better than the old scheme. Also some of the results and calculations need further explanation.

Dear referee,

Thank you for your thorough review of our preprint. Your comments helped us to not only improve the shape of the paper but also to clarify our key results. In particular, as recommended by your review, we refined the evaluation of our spectral fluxes with the help of additional data coming from the CLAAS-3 dataset of the EUMETSAT Satellite Application Facility on Climate Monitoring (CM SAF). Using a binary cloudiness mask from this dataset, we could better assess the cloudiness of MAR and its impact on our spectral shortwave flux biases. As a result, our evaluation of spectral shortwave fluxes (end of Section 5.2) should be much more complete than initially.

While we did not change our key results with respect to gridded products (Section 4), the remarks regarding the differences between our nine experiments guided us in revising the text commenting the results. The revised paper now provides more precise comments regarding the mean shortwave flux differences that we obtained in our nine experiments. In particular, we stressed the importance to bring these differences as close to zero as possible in the context of our research. Indeed, one of our motivations for including the ecRad radiation scheme in the MAR model was to be able to produce accurate spectral shortwave fluxes. Without our tuning efforts, MAR would consistently overestimate shortwave radiation on a decadal basis by several percents (up to 10%) with respect to the mean shortwave flux over Belgium for the whole 2011–2020 decade. We are confident our revised paper provides now more compelling conclusions and messages with respect to our results.

It should be noted, though, that we used new software in the meantime to produce our grid-wide statistics and some of our figures (notably the maps). We also corrected a minor issue with our spectral evaluation, which was that the MAR time series was not exactly centered on Uccle due to a minor indexing issue at the time of extracting the spectral flux time series. As a result, some of the numbers that were provided in the previous version may have slightly changed in the revised paper, but they remain very close to what we presented initially.

Finally, we did as much as we could to improve the language and readability throughout the paper, as recommended. As the final manuscript is quite long, we could only proof-read (for language) the sections that changed the most between the previous manuscript and the revision.

You can find below detailed responses to your various comments regarding the content. For comments regarding the readability and language, we provide a response only when we partially applied the recommended changes.

Best regards,

Jean-François Grailet

**Responses to separate comments from RC1**

**Opening remarks:**

- For suggested changes that were applied directly without any change, no response is provided.
- Likewise, when a suggestion regarding language was applied beyond the scope of the lines and paragraphs mentioned in the review, no response is provided. We assume the PDF highlighting the changes should be enough in this regard.
- On a side note, we decided to add David Bolsée (who provided us with the spectral observations at Uccle) as a co-author. We formerly only mentioned him in the acknowlegements. We also applied a few changes to section 5 based on his feedback.

> Figure 1: Long dashed versus short dashed - I'd suggest you make this clearer. May be better to apply colour instead.

We updated **Figure 1** to address this comment as follows. Rather than using long and short dashed arrows, we decided to use continuous arrows for both inputs and outputs and dashed arrows for ecRad internal data structures.

Initially, the dashed arrows coming out of the *Thermodynamics* box were meant to visually set this box apart from the other input boxes. Indeed, the description of temperature and pressure are taken into account by all ecRad sub-components, while all other boxes feed only one ecRad component at a time.

However, after reviewing the figure, it appeared to us that the multiple arrows coming from the *Thermodynamics* box were not ambiguous, so we made all arrows coming from inputs and going to outputs continuous to ensure they would not be mixed up with the ecRad internal data structures (symbolized by dashed, red annotated arrows).

> L133-134: that manages spectral bands extending over a few dozens of nanometres, e.g. 25 nm – slightly unclear

To make this part clearer while staying consistent with the cited work, the example was re-expressed as "*from 400 to 425 nm*" (**L133** in the revised manuscript).

> L139: By swapping RRTM-G with high resolution ecCKD gas-optics models, — what's the computational cost?

In an equivalent setting, ecCKD models are more efficient, because they usually define less spectral intervals (*g-points*) than RRTM-G. These g-points are meant to model the spectral variation of gas absorption within each spectral band, and have a direct impact on performance. Even if ecCKD define more spectral bands than RRTM-G, they typically use less spectral intervals, rendering them more efficient than the latter. This was already mentioned in the final paragraph of Section 4.1, but to already mention this fact in this part of the text, we re-worked the beginning of the final paragraph of Section 2.2 as follows, starting from **L138** in the revised manuscript. You can also find the revised text below.

*Using ecCKD to increase the spectral resolution also comes at a smaller computational cost. Like RRTM-G, ecCKD models define spectral intervals within spectral bands, or "g-points", to model the spectral variation of gas absorption within bands. The number of such* g-points, *which has a direct impact on performance, is typically lower in ecCKD models than in RRTM-G (for both shortwave and longwave radiation), even when the former define more spectral bands. As a result, by swapping RRTM-G with high resolution ecCKD gas-optics models, ecRad can easily output equally high resolution spectral shortwave fluxes [. . .]*

> L158: The number of aerosol species — any limitations on this??

As of now, it is possible to define up to 256 aerosol species in ecRad. This detail was added in the revised text, as follows (cf. **L162** in the revised manuscript):

*The number of aerosol species taken into account by ecRad can also be tuned, users being able to define up to 256 different species.*

> L160-175: Not sure if all of these details are needed here - perhaps could go in an Appendix either? It somewhat disrupts the flow of the paper.

We believe we should keep these details, which we already simplified many times while writing the paper, for at least two reasons. First, they represent a departure from how MAR used to prepare greenhouse gas mixing ratios, and current MAR users, at the very least, will be interested in getting the big picture of this update.

Second, people who are familiar with the forcings themselves may be curious about how we adapted them to the MAR grid given the large resolution difference. For reminders, a single air column from the greenhouse gas climatologies cover around 3° with respect to latitude, which is more than 300 km. The resolution of MAR, on the other hand, typically ranges from 5 km (sometimes less, e.g., 2.5 km) to 25 km.

> L176: CAMS aerosol specification, consisting of 11 hydrophilic or hydrophobic aerosol species — are there not 14 species?

We use the 3-D monthly aerosol forcings provided on the ecRad wiki (https://confluence.ecmwf.int/display/ECRAD/), base on the 2020 work by Bozzo et al. (https://gmd.copernicus.org/articles/13/1007/2020/gmd-13-1007-2020.pdf), which provides only 11 species. It is worth noting, though, that the IFS routine that initializes the ecRad radiation scheme (whose code helped us update MAR to v3.14) defines one additional species, stratopheric sulphate, which we eventually removed since we lacked the data to model it.

> Para around L200 - are you saying the the vertical grid is different only for the rad calculations?

Yes, it is. To make this clearer in the text, we added a sentence just before the one at L200 (original text), i.e., "*This means the MAR model . . . [. . . ]*". The sentence (**L203** in the revised manuscript) is given below. We further emphasized this fact in other parts of the paper, such as Section 4.3.

*In other words, the MAR grid is extended with stratospheric pressure layers only during radiation calculations.*

> L228: an underestimated cloudiness, the radiation scheme also requiring cloud fraction values for each grid cell among its inputs. — does not read well.

We simplified this part as given below (**L233** in the revised manuscript), assuming the reader will understand *de facto* that cloud fraction values are part of the inputs of the radiation scheme.

*[. . . ] an underestimated cloudiness, the radiation scheme requiring cloud fraction values for each grid cell.*

> Figure 4 caption – too long a sentence - confusing to read.

We split the caption of **Figure 4** into multiple sentences, some of which were reformulated. You can find the new caption below.

*Comparison of the ozone volume mixing ratios fitted to a MAR grid over Belgium, before and after adding the extra pressure layers. The mixing ratios are vertically integrated (g~m^-2) and averaged on a monthly basis. The concentrations in the extra layers are adjusted to those found at 1 hPa, 10 hPa and 30 hPa. Based on the outputs of a sample MAR v3.14 run over Belgium in 2019.*

> ut are good at predicting the total cloud cover, which is currently the most important requirement for MAR regarding cloudiness, given that research involving the MAR model focuses mostly on (near-)surface processes, — I don't agree - cloud condensate has a huge impact on the surface variable, more so than cloud cover - compare a high thin cloud to a low thick cloud, both having 8 octa.

Modification to the microphysics scheme to change the water content in the model would be a major undertaking with significant impacts on precipitation rates and latent heat release, in addition to radiation. As with any model, we are always looking to make improvements to each of the components in MAR, including its microphysics scheme, but this is out of scope for the present study. By contrast, the cloud fraction parameterization is used only by the radiation scheme so is a reasonable place to focus our attention in a paper about updating the radiation scheme. Modification to the microphysics scheme will have to wait until a future paper.

As an aside, the reviewer asserts that cloud condensate is more important than cloud fraction, but this is not true for the same fractional change in the two variables. We have used the offline ecRad package with the 361 profiles in the package's "era5slice.nc" file to test the impact of reducing cloud fraction by 10% of its default value versus reducing the liquid and ice mixing ratios by 10%. The impact on shortwave fluxes is similar, but the impact on longwave fluxes is around five times larger from changing cloud fraction than from changing mixing ratios.

> L251: (i.e., droplets, ice crystals and snowflakes), — is graupel and hail accounted for in these?

In the MAR model, graupel and hail are assimilated with snow. We therefore revised the text in parentheses as follows: "*[. . . ] (i.e., droplets, ice crystals, snowflakes, hail, etc.)*" (cf. **L256**).

> State disadvantages of ——- such as more advanced diagnostic parameterizations (Weverberg et al., 2021b, a) or prognostic solutions (Tompkins, 2002), both requiring more implementation work. The inclusion of these more complex solutions in the MAR model is left for future work.

The original text already states the main disadvantage in the context of our research. We did not try to include these solutions because both methods are whole schemes on their own (i.e., it would justify writing additional subroutine(s) to call before calling the radiation scheme), while the parameterizations we included for this research could be implemented by just adding a few instructions to the MAR code. Without the time constraint, we would certainly have tried to include one of these. Finally, we also prefer not to dive too deep into the merits of each method with respect to each other, as it would require several sentences to provide a satisfactory summary, which is not much relevant to our present study.

> L257 - parametrization is the correct spelling I believe

We decided to stick to our current spelling (*parameterization*) as they are both used in the titles and texts of the research papers by Sundqvist (1989) and Xu and Randall (1996).

> L265 - sentence a bit long and clunky. Are you saying your can't use the LW scattering with Tegen - unclear from the sentence.

In the ecRad version we used, it is indeed not possible to simulate the longwave scattering effect of aerosols if ecRad is configured to use the Tegen climatology. To clarify this, we rewrote this part as follows (cf. **L269** in the revised manuscript).

*Likewise, MAR always prepares its aerosol forcings for ecRad with a monthly aerosol climatology compliant with the CAMS aerosol specification (Flemming et al., 2017) rather than the old aerosol climatology of Tegen (1997). With this configuration, MAR can also enable ecRad to simulate the longwave scattering effect of aerosols, an additional process ecRad is unable to perform with the Tegen climatology.*

> I think default settings should be included in the paper rather than referring to a wiki, which can be updated at any stage.

We agree with this observation and decided to include a new Table (**Table 2**) to provide the main options of ecRad configured in MAR, including for our sensitivity experiments in Section 4. This said, please note the ecRad documentation, which the revised text still cites, enumerates many more parameters than we mentioned in both our text and the new Table. Hopefully, from our experience, such parameters have a negligible impact on the both the accuracy and the performance of ecRad within MAR.

> The phrase simplicity's sake appears many times - not good use of English.

We removed both instances in the first paragraph of Section 3.4, as conveying the idea of "simplicity" may sound strange in this context. For the first instance (second sentence of the first paragraph, **L266** in the revised manuscript), we opted for the following.

*To quickly establish a default configuration for MAR, ecRad was configured in MAR with its default or most modern options [. . . ]*

For the second instance, in the part about parallelization, we used the following sentence (**L273** in the revised manuscript).

*While not a option of ecRrad, MAR also re-uses the parallelization strategy of the stand-alone implementation of ecRrad [. . . ]*

> L271: Again a reason to have a table of the default settings - unclear from reading whether TripleClouds is the default or you have chosen a non-default option. Change McICA to McICA scheme.

By default, ecRad runs with the McICA scheme. To clarify, this sentence was edited to mention the fact that ecRad uses McICA by default, which means Tripleclouds is our own choice, as highlighted in the new Table 2. You can find the new version below (**L276** in the revised manuscript).

*Among the three available solvers in ecRad, the TripleClouds scheme (Shonk and Hogan, 2008) was picked as the default solver to be used by ecRad in MAR, the McICA scheme (Pincus et al., 2003) being the default solver in ecRad.*

> L300: Second, we evaluated. . . . . also the sentence is too long.

We split the sentence in two as suggested. You can find the new version below (**L307** in the revised manuscript).

*Second, we evaluated whether or not a well tuned MAR v3.14 could produce better radiative fluxes with ecRad than with its previous configuration (i.e. still using the Morcrette scheme). In the process, we also assessed if there was any negative impact on other MAR variables.*

Please note we moved the bit about the heat flux tuning mechanism farther in the revised text, at the end of the paragraph (now starting at **L315**) that cited the multiple references on MAR radiative flux biases. This bit has also been reformulated as follows.

*By including and tuning ecRad in MAR, we also aim at ensuring MAR no longer needs such a [heat flux tuning] mechanism.*

> L304/305: respectively without and with a heat fluxes tuning mechanism inherited —- with and without a heat flux tuning mechanism

We deliberately used *without and with*, as they refer respectively to our experiments M1 and M2. M1 is the one using the Morcrette scheme without the heat flux correction. To switch to *with and without*, we would need to advertise M1 and M2 in reverse order, which would not feel natural.

On a side note, we split the sentence in two to improve readability, as follows (**L310** in the revised manuscript).

*Our first two experiments, M1 and M2, consisted of running MAR v3.14 with the Morcrette scheme, respectively without and with a heat flux tuning mechanism inherited from previous versions of MAR. This tuning mechanism is applied right after heat fluxes have been deduced from the outputs of Morcrette.*

> Re ecCKD - which version did you use? The new sped-up one by Peter Ukkonen (refactored) or the old one?

For both ecRad and ecCKD (we used the latter to tune some models for the spectral features), we used the source code available on the respective ECMWF GitHub repositories. We did not use the refactored version by Peter Ukkonen.

> L312: Starting with . . . ., were added —- explain this better - have you changed the number of model levels in the runs?

As mentioned earlier, stratospheric pressure layers are only added during radiation calculations. As such, the number of vertical levels in MAR stay the same throughout the experiments (24 pressure layers in sigma coordinates). From E2 to E7, however, we always used the 3 additional stratospheric pressure layers during radiation calls, as suggested by Table 2 (in the initial text).

> from E3 to E6) test the — rm from . . . tested the. L314: parametrization . . . . for the fw parameter of ecRad —- of the fw parameter. L315: tense. ecCKD gas-optics models —- model not plural

We applied most of the suggested changes and changed tense in other places of the same paragraph to stay consistent. We however kept model**s** in plural: ecCKD models are built either for the shortwave or the longwave, so we are technically using two such models. On top of that, the ecCKD shortwave model we used was tuned specifically for our research: prior to our research, the highest resolution shortwave model overestimated ozone absorption in the UV range. The longwave ecCKD model we used, however, was already available as soon as 2022 on the ECMWF GitHub repositories.

We however kept referring to RRTM-G in the singular form, despite that there are technically two schemes (again, for shortwave and longwave), to avoid making too many changes to the paper, but also to avoid confusing the reader (referring to multiple RRTM-G schemes may imply other things, such as multiple versions). We nevertheless clarified the paragraph starting at L314 in the initial text (now **L322**) as follows.

*The final ecRad experiment, E7, reused the configuration of E4 but swapped the classical RRTM-G scheme **for both shortwave and longwave** with high resolution ecCKD gas-optics models (Hogan and Matricardi, 2022), with two goals in mind.*

L336 - sentence way too long and clunky. L339: due to a lack of a gridded — due to the lack L345: only with a different projection system – only on a different projection. MSG satellite – singular

We rephrased the (long) sentence starting from L336 (initial manuscript) as follows (now starting at **L343** in the revision).

*The daily mean downward longwave fluxes, i.e. the last of the four assessed MAR physical variables, are directly compared to MSG daily DSLF dataset (MDIDSLF). This gridded product provides surface downward longwave fluxes as recorded by the successive MSG satellites of EUMETSAT (Trigo et al., 2011a) at 0.05 degree latitude-longitude resolution over a region covering Europe, Africa, part of South America and the Middle-East. We chose to use this product due to the lack of a gridded RMIB longwave product equivalent to the RMIB shortwave product.*

We kept the plural for the MSG satellites because there are actually successive satellites for our period of evaluation: 2011-2013 was captured by MSG-2, 2014–2018 by MSG-3, and the remaining 2019–2020 by MSG-4.

L341: relative error below 10% compared — where does the 10% come from, should it be 20?

This 10% threshold does not come from us but from the report we cited a bit before, at L340 of the initial text.

L346: degrees

In this case, we use *degree* as a unit. Therefore, we prefer using the singular.

L348: which can be visualized on the grids of the data products, have been both saved and averaged, —- don't think you need to mention they have been saved etc.

We simplified this sentence as follows.

*The resulting 2-D statistics have been averaged with respects to the grid and are provided in Table 4.* (**N.B.:** table number has changed from 3 to 4)

L355: Paragraph not well written. Language vague - perhaps include more of the values. Not conclusive about what's happening in each experiment. More detail needed in order to distinguish the options. Some of the biases quoted are surely less than the bias on the satellite product in the first place. Mention of the extra levels here - should that be described much earlier in the paper? Looking at Table 3 - diffs are not major and within range of uncertainty in the obs. L365 - the diffs are so small, I would say you can't really draw such conclusions from them. L365-380: This discussion is a bit weak I think - I don't think too much weight can be put on the small differences. L385-395: explain these results a bit better and why you are confident in them? They are not all stat significant though so difficult to draw such conclusions. Are you have if ecRad is comparable to Morcrette, so as long as it's comparable it's OK, and you switch to the more advanced scheme.

To address these (related) comments, we first slightly toned down our conclusions regarding the improvements brought by ecRad. We agree with the observation that the changes in our temperature and precipitation statistics are not significant. We therefore prefer only stating that we have similar statistics across all experiments for temperature and precipitation, regardless of the radiation scheme. We also slightly changed our abstract to no longer state MAR v3.14 "*outperforms*" former versions. Instead, we mention that MAR v3.14 is "*better equipped*" than before to achieve balanced radiative fluxes, i.e., radiative fluxes that are close to zero on a (pluri)annual basis and on a seasonal basis.

Indeed, we are confident the evolution of the radiative flux differences across our experiments, and in particular, from E1 to E4, remain noteworthy in the context of our research. Through our adjustments, we went from a mean decadal shortwave flux difference of sligthly more than +12 W mˆ-2 (E1, i.e., ecRad used "*off the shelf*") to near zero (E4 or E7). Given that the average shortwave flux for the 2011–2020 decade and the whole Belgium territory is about 123 W mˆ-2, a mean shortwave flux difference (averaged on the grid) of +12 W mˆ-2 amounts to a relative change of almost +10%. This is while considering a domain-averaged

difference, as the differences for individual grid cells can be even larger: from less than -10 W mˆ-2 with M2 in the north of Belgium (hinted at by Figure 5) to almost +20 W/m² in the south with E1, both on a decadal basis.

Considering that one of the goals of our present research is to produce spectral shortwave fluxes through ecRad, in order to force other models or to envision new applications (like UV index computation), using an ecRad configuration with almost +10% shortwave radiation on average does not sound acceptable to us. Moreover, as we have demonstrated in both the initial text and the revised text of Section 5, bringing the mean decadal shortwave flux difference to near zero is not enough to mitigate positive spectral shortwave biases.

We therefore significantly revised **Section 4.2** (including to address the next comment, cf. below), both to put the shortwave flux differences into perspective (i.e. with respect to the average value in the RMIB shortwave product) and to stress the fact that we want to bring the mean radiative flux differences as close to zero as possible, such that using MAR v3.14 to produce spectral shortwave fluxes becomes desirable.

> L370 + paragraph: Can you justify more why you can use fw=0.5? At higher res the inhomogeneity should be closer to 1?

Our decision to use f_w=0.5 results both from experience and from the content of the study by Shonk et al. (2010) which we refer to multiple times in our paper, most notably in Section 3.4 (default configuration) and Section 4.2 (where the aforementioned line appears).

On the one hand, in Table 1 from Shonk et al., the mean f value above land derived from midlatitude and global datasets over land typically leans towards lower values of the $0.75 \pm 0.18$ recommended range, with resolution ranging from 5 km to 0.45 km. In the same table, higher values for f_w (i.e., at the very least above 0.65) are associated with datasets covering a specific season (i.e., winter or summer), the ocean or ice clouds only.

On the other hand, lowering f_w in ecRad results in practice in bringing the variability of the water content of clouds closer to the mean. This is illustrated in Figure 1 from Shonk et al. (2010). The practical consequence on ecRad is an enhanced cloudiness which impacts the radiative balance.

We revised our paragraph commenting the results of E5 and E6 (**L381** in the revised manuscript) to better stress both the motivations of such experiments and what they mean for MAR users. I.e., while using 0.75 for f_w in ecRad is acceptable for Belgium, lowering the value of such parameter may help MAR users to adjust the radiative balance over other regions.

> Fig 5/6: Might be good to show areal plots for all experiments - nicer than the summary stats and can also see any geographical variation.

While this suggestion is well intended, we believe such an addition would not bring much new information to readers. The geographical repartition of the shortwave flux differences is roughly the same in all ecRad experiments. In particular, these differences are practically the same between E4 and E7, and showing a map similar to Figure 6 for E4 would visually amount to duplicating Figure 6. Moreover, for all figures to be easily comparable, we would need to extend our color scale to [-20,+20] W mˆ-2 to make sure the E1 and E2 differences can be gauged with respect to other experiments, at the cost of making the differences between M1, M2 and E7 (and E4) harder to grasp.

This said, we revised the end of Section 4.2 to inform readers about the aforementioned similarities between ecRad experiments (and in particular, E4 and E7 maps being mostly the same), starting at **L408** in the revised manuscript.

> L400 etc: I don't think you should argue that you're happy to have both + and - biases compared to all in one direction.

We annotated **Figure 7** to underline the grid-wide mean, as we target a grid-wide of near zero, as discussed throughout the revised Section 4.2.

> Table 5: Might be good to include all ecRad expts here and not just 3. Perhaps include expt number in the table to avoid having to decribe again in the text

We updated **Table 6** (formerly Table 5) to provide the experiment numbers as suggested. However, we still use only 3 experiments, as they are representative of the computational costs of all seven ecRad experiments. To make this clearer, we added a column in Table 6 to advertise the experiments represented by all three assessed configurations. In practice, only one accounts for multiple configurations. We revised the beginning of Section 4.3 (starting at **L421** in the revised manuscript) to account for these changes as well.

> L415 - have you (or can you) test the refactored ecCKD?

As mentioned earlier in this response, we did not use the refactored ecCKD by Peter Ukkonen.

> L424: Can you explain this - longer time elapsed but yet ecRad more efficient L423-430 - written confusingly. Please elaborate more on the numbers and your conclusions re speed of ecRad. I don't follow how ecRad is much slower but overall runtime more or less similar. What causes that? L433: You mention the 26% but have not put it in context compared to the other expts.

To address all three comments, we revised the last two paragraphs of Section 4.3, now written as three paragraphs (starting from **L439** in the revised manuscript). The minor changes in overall times despite ecRad calls being longer can be explained by the fact that all radiation scheme calls lasted less than one second in all experiments. The revised text provides this additional detail. At the scale of the whole MAR execution, the randomness of other operations, such as I/O calls to read or write on disk, had a greater impact on MAR performance in our experiments.

Moreover, the revised paragraphs also detail more exhaustively the resolution difference between Morcrette and ecRad, which is most likely one of the main factors while even the best ecRad configuration (w.r.t. performance) remains slower than the Morcrette scheme.

> L458: Define BISA

We provided again the full name of the Royal Belgian Institute of Space Aeronomy in this place (**L477** in the revised manuscript). Please note, however, that we no longer use "BISA" in the updated manuscript, as David Bolsée (now a co-author) informed us all publications involving the Royal Belgian Institute of Space Aeronomy should refer to it as BIRA-IASB (which stands for *Koninklijk Belgisch Instituut voor Ruimte-Aeronomie – Institut royal d'Aéronomie Spatiale de Belgique*).

> L480: diurnal measurements — do you mean daylight measurements?

Yes. We replaced *diurnal* with *daylight* just in case. However, we did not use this word at L480 but at L498.

> L485 Paragraph a bit confusing - write more clearly

Given the line mismatch above, we were not entirely sure which paragraph this comment referred to, as the paragraph starting at L481 is technical. However, in hindsight, the paragraph detailing our spectral bands at the end of Section 5.1, starting at L474 (of the initial text), deserved an improvement. We therefore revised this paragraph (at **L494** in the revised manuscript) to clarify our spectral bands and their relationship with our ecCKD shortwave model. We hope this revision will be welcomed.

> Fig 7: Can you comment on the overestimation in MAR compared to OBS across all bands? I know it's small but seems consistent. Fig 8: Would it be better to filter by clear days to try to figure out what's causing the difference? The cloud days are more complex as you will always have cloud mismatches. Have you looked at any clear sky indices for MAR vs Obs over the time period? Clear sky index would clearly show whether you are lacking cloud in certain situations.

To address these comments, we searched for additional data regarding cloudy days above Uccle and found a hourly binary cloudiness mask in the CLAAS-3 dataset by the EUMETSAT Satellite Application Facility on Climate Monitoring (CM SAF). To approximate a daily mean cloud cover that we could compare to our MAR cloud cover variable (coming from E7), we computed daily averages of the CM SAF cloudiness mask.

Using our cloud cover time series, we isolated days that were cloudy in both MAR and the CM SAF data, and days that were cloudy only in the former. We then updated Figure 8 to provide not one, but two spectrum figures, where the comparable time steps have been reduced respectively to cloudy days in both MAR and CM SAF, and cloudy days according to CM SAF only (thus, partly cloudy or clear in MAR). We are confident the resulting figures better support our initial conclusions regarding an underestimated cloudiness in MAR.

Moreover, we significantly revised our text from L515 to L529 (initial manuscript) at the end of Section 5.2 (initially a single paragraph, revised into three; from **L536** in the revised manuscript), both to explain our updated methodology to assess MAR cloudiness and to give additional details about our results. In particular, by counting how many time steps were processed for both spectra in Figure 8 per month, we discovered the cloudy days missed by MAR mostly happened during the summer months, and July especially. These additional results both detail when MAR tend to underestimate cloudiness and explain the differences of magnitude between both spectra in the updated Figure 8.

> L560 and before/after: Perhaps all the calculations should go in an appendix so as not to distract from the results and discussion.

We considered this suggestion, but eventually decided to keep the initial text. The formulas given in the paragraph from L554 to L567 (initial text) are indeed a translation for the MAR spectral bands of the (continuous) definition of the UV index. We believe readers will wish to know how we adapted the UV index definition to MAR spectral bands, considering that such bands have a non negligible resolution difference w.r.t. the Uccle observations (even when integrated on 1 nm wide bands).

> L575 - can you comment on why MAR can't capture maxima? Is it the resolution? How often is the radiation scheme called - has this been experimented with? OK I see you answered later but perhaps add comment to the section.

We added a sentence at the very end of Section 5.3 (**L559** in the revised manuscript) to already hint the discussion with respect to UV indices.

*This slight underestimation from MAR may be a consequence of mismatch between the total ozone during the observations period and the total ozone in MAR (cf. Sect. 5.4).*

**Response to Referee 2 (RC2)**

> The authors have done a nice job with implementing the newest version of ecRad in the MAR regional atmospheric model. The paper describes the testing of this concisely and gives an nice use case with computing the UV index.

Dear referee,

Thank you for reviewing our manuscript. We took your comments carefully into consideration and made several modifications as a result, mostly impacting the shape of the paper.

We also adapted the terminology for our evaluation (Section 4). Based on your comments, we decided to drop "*root mean square error*" and "*bias*" entirely in favor of "*root mean square deviation*" (RMSD) and "*difference*". This updated terminology should be more appropriate than the one we used in our previous manuscript.

You can find detailed responses to your comments below.

Best regards,

Jean-François Grailet

**Responses to separate comments from RC2**

**Opening remarks:**

- For suggested changes that were applied directly without any change, no response is provided.

- On a side note, we decided to add David Bolsée (who provided us with the spectral observations at Uccle) as a co-author. We formerly only mentioned him in the acknowlegements. We also applied a few changes to section 5 based on his feedback.

  - Figure 1: The available cloud optics schemes for liquid and ice clouds should be listed.

We updated **Figure 1** to add these schemes in the flow-chart. To comply with a comment from Referee 1, we also added a Table (**Table 2**) in Section 3.4 to enumerate the options we picked to configure ecRad in MAR v3.14. For the sake of consistency, all components from Figure 1 are mentioned in this table too.

  - Line 141: I don't understand the text in the parentheses. Can you reformulate this?

We decided to remove the bit in parentheses and add a full sentence to the revised text, roughly at the same place, to clarify what the initial text meant. Please note, however, that the whole paragraph where the initial text in parentheses appears has been significantly revised to address another comment from Referee 1 (about the computational cost of swapping RRTM-G with ecCKD models).

You can find the revised text below, which is found at **L138** of the revised manuscript (new final paragraph of Section 2.2).

*Using ecCKD to increase the spectral resolution also comes at a smaller computational cost. Like RRTM-G, ecCKD models define spectral intervals within spectral bands, or "g-points", to model the spectral variation of gas absorption within bands. The number of such* g-points, *which has a direct impact on performance, is typically lower in ecCKD models than in RRTM-G (for both shortwave and longwave radiation), even when the former define more spectral bands. As a result, by swapping RRTM-G with high resolution ecCKD gas-optics models, ecRad can easily output equally high resolution spectral shortwave fluxes, spanning over one or several dozens of nm, assuming the bands requested by the user are not finer in resolution. In practice, requesting finer spectral bands than defined in the ecCKD shortwave gas-optics model used by ecRad results in spectral fluxes that are proportional to the encompassing ecCKD spectral band.*

  - Section 3.1: "Updated greenhouse gas and aerosol forcings". The SSP scenarios include aerosol scenarios. Did you include these? If not, then explain why not! The direct, and in particularly the indirect, effects of aerosols are of similar magnitude as the greenhouse gases regarding contemporary climate change (Forster et al. 2024).

The MAR model currently does not model the interactions between clouds and aerosols, just like the ECMWF model. These interactions or time-evolving mixing ratios have little impact for weather forecasting, and while our present paper did not directly mention weather forecasting applications of MAR, we do use MAR (including v3.14) to predict the weather up to 10 days in the future, using GFS outputs as lateral boundary forcings.

This said, there is ongoing research in Grenoble to run MAR v3.14 with time-evolving aerosol mixing ratios based on IPCC scenarios, rather than relying entirely on the CAMS climatology as we did for this paper.

  - Table 3: Are the RMIB gridded precipitation data based on radar and rain gauge data, on interpolated rain gauge data, or on modelled precipitation data? If it is based on one of the latter two, I would suggest to refer to the resulting statistics as "Difference" and "RMSD".
  - Table 3: For the LW data "Bias" has been exchanged with "Difference" - presumably due to the more uncertain nature of the MSG LW radiation data. To be consistent, "RMSE" should here be replaced with "RMSD".

We will address both comments at once, as they are closely related. To answer the first comment, the RMIB precipitation product is indeed based on interpolated rain gauge data. In fact, all RMIB products were built by interpolating observations recorded by a dozen of weather stations scattered across Belgium, though the shortwave product has the advantage of being combined with satellite data for greather accuracy.

In all cases, only specific grid points from RMIB products may be considered as a form of "*groundtruth*", with the longwave product consisting exclusively of satellite data, thus with an added uncertainty compared to surface measurements. Therefore, to address both comments thoroughly, we decided to generalize the use of "*difference*" in place of "*bias*", and "*RMSD*" in place of "*RMSE*" in both **Table 4** and **Table 5** (formerly

Table 3 and 4, respectively) and more generally throughout Section 4.2. However, we kept using "$bias$" and "$RMSE$" for the evaluation of our spectral fluxes (Section 5.2), where we do rely on ground observations.

---

## Author Response (AR3)

**Author's response (third revision of MAR v3.14 manuscript)**

Dear Dr. Mann,

Thank you for your comments on our revised manuscript and for your suggestions to further improve the language. Since the suggested revisions were straightforward, I applied all of them, though I used a slightly different wording for two of them (see under *Additional comments on revisions*). I also made additional minor revisions to further improve language (see under *Additional revisions*).

Regarding the spacing issue that occurred with acronyms in the previous TC-manuscript, I must admit that I didn't spot it at the time of submitting the previous revision. I did use latexdiff to quickly generate the track-changes file, though I had to reduce the font size in some tables myself to make sure they could be reviewed to their full extent.

After looking into the problem, it turned out latexdiff was bothered by my acronym commands. For each acronym I use in a LaTeX document, I have the habit to define a LaTeX command instead of writing it in plain text so I don't have to change the whole manuscript if I have to change the capitalization or spelling of the name of a tool, institution, etc. Thus, each time latexdiff found a change containing such acronym command(s), it ended then re-started its own commands around them, and this cancelled a white space that followed each acronym command. You can find an example below.

```
\DIFaddbegin \DIFadd{By including and tuning }\ecrad \DIFadd{in }\mar\DIFadd{, we also
aim at ensuring }\mar \DIFadd{no longer needs such a mechanism.
}\DIFaddend
```

A quick though manual solution I found is to edit the LaTeX code around acronyms encompassed by changes to remove closure/restart around them.

```
\DIFaddbegin \DIFadd{By including and tuning \ecrad in \mar, we also
aim at ensuring \mar no longer needs such a mechanism.
}\DIFaddend
```

The new TC-manuscript should no longer feature this spacing issue around acronyms, which is absent from the final manuscript.

Thank you again for your feedback and for overseeing the peer-review process of our paper !

Best regards,

Jean-François Grailet

**Additional comments on revisions**

> 2) Lines 438-439: The sentence here begins "On the one hand, . . . .", but there is no corresponding "on the other hand" to follow later. The sentence seems fine to simply begin "These seasonal statistics" rather than "On the one hand, these seasonal statistics", and please re-word for that.

There was actually a "*on the other hand*" a bit farther in the same paragraph (at L442 in the previous TC-manuscript), but there was an intermediate sentence ("*In particular, the correlation coefficients [. . .]*") to detail the observed trends. To improve the text, I decided to remove the "*on the one hand*" in the text above (L395 in the new TC-manuscript) and to replace the "*on the other hand*" found a bit later by "*however*". The new text is given below (L397 in the new TC-manuscript).

*[. . .] during this season, M1 and M2 provide equivalent or better results.* **However,** *the seasonal statistics for M1 and M2 show non-negligible seasonal disparities regardless of tuning the heat fluxes.*

> 3) Line 482 – The "lie in" in the wording "The differences between these configurations lie in the choice of the cloud fraction parameterization. . . " seems somehow not quite right. Please replace "line in" with "are from", as this would be better wording.

I rephrased this sentence as suggested, but using a slightly different wording, as shown below (L426 in the new TC-manuscript).

***These configurations only differ in*** *the choice of the cloud fraction parameterization and the f_w parameter.*

**Additional revisions**

**Reworded "on the one hand. . . on the other hand. . . " (x2)**

Since I used a few times "*on the one hand. . . on the other hand*" several sentences apart a few more times in the previous TC-manuscript, I decided to remove and/or reword these occurrences to be consistent with the revision suggested for Lines 438–439 (previous TC-manuscript). The changes are listed below, using the line numbers of the new TC-manuscript.

- L357: "*Table 4 demonstrates*  *that all experiments performed well with respect to the RMIB products* [. . . ]".
- L361: "*The average statistics for radiative fluxes*  *are more contrasted.*".
- L624: "[. . . ] *to assess the current limits of MAR v3.14.*  *Table 5 from Sect. 4.2* [. . . ]".
- L633: "***Another shortcoming of MAR v3.14,*** *hinted by Sect. 3.1,* ***is the limited temporal resolution of greenhouse gas and aerosol concentrations,*** *due to the initial forcings consisting of monthly means*".

**Too many "in particular"**

While revising Section 4, I realized I used a bit too many times "*in particular*" throughout the manuscript, with two occurrences in at least two paragraphs of Section 4.2. I decided to improve this by rewording some occurrences or removing them altogether when they felt unnecessary. These changes are listed below, using the line numbers of the new TC-manuscript.

- L362: "***For example****, E1, which use****d*** *none of the adjustments described in Sect. 3, yielded a mean shortwave flux difference of +12 W m^-2*".
- L370: removal of the sentence "*In particular, the mean differences are continuously decreasing for both shortwave and longwave radiation from E1 to E4*": on second thought, it did not bring much with respect to the previous sentence.
- L404: removal of the sentence "*In particular, all seasonal mean differences are below 5 W m^-2 in absolute value with the exception of the summer*": it added an anecdotal observation with respect to the previous comments in the same paragraph.
- L626: " *as highlighted in Sect. 5.2 (with the help of Fig. 8), MAR struggles to correctly predict cloud cover during the summer*".
- L677: "[. . . ] *to produce new forcings for other computer models, and* ***especially*** *those requiring spectral shortwave fluxes in the photosynthetically active region*".